# REWARD-FREE EXPLORATION BY CONDITIONAL DIVERGENCE MAXIMIZATION

## ABSTRACT

We propose maximum conditional divergence (MaxCondDiv), a new curiosity-driven exploration strategy that encourages the agent to learn in the absence of extrinsic rewards, effectively separating exploration from exploitation. Our central idea is to define *curiosity* as the divergence between the agent's estimation of the transition probability between the next state given current state-action pairs (i.e., $\mathbb{P}(\mathbf{s}_{t+1}|\mathbf{s}_t, \mathbf{a}_t)$) in two adjacent trajectory fractions. Distinct to other recent intrisically motivated exploration approaches that usually incur complex models in their learning procedures, our exploration is model-free and explicitly estimates this divergence from multivariate continuous observations, thanks to the favorable properties of the Cauchy-Schwarz divergence. Therefore, MaxCondDiv is less computationally expensive and reduces internal model selection bias. We establish a connection between the MaxCondDiv and the famed maximum entropy (MaxEnt) exploration, and observe that our MaxCondDiv achieves wider exploration range and faster convergence. Our exploration also encourages the agent to acquire intricate skills in a fully reward-free environment.

## 1 INTRODUCTION

Over the past few years, Reinforcement Learning (RL) has achieved remarkable success in addressing challenges in fields like robotics (Mnih et al., 2015) and games (Silver et al., 2016). Nonetheless, RL's practical applications in real-world scenarios are still restricted due to the high variability and lack of user control on the availability of dense rewards, which are critical to the timeliness and success of RL. To counteract this shortcoming, intrinsically motivated exploration (Amin et al., 2021) has been put forth to encourage the agent to explore unknown states in the absence of extrinsic rewards, by offering an internal motivation, such as diversity (Eysenbach et al., 2019), novelty (Ostrovski et al., 2017; Tao et al., 2020), curiosity (Pathak et al., 2017; Burda et al., 2019).

Existing intrinsically motivated exploration approaches can be roughly divided into two categories with distinct goals (Amin et al., 2021): the *space coverage* approaches encourage an agent to visit more unexplored states or state-action pairs in a shorter amount of time; whereas the *curiosity*-driven approaches seek to explore areas where the agent's prediction on next state given current state-action pairs (i.e., $\mathbb{P}(\mathbf{s}_{t+1}|\mathbf{s}_t, \mathbf{a}_t)$) has high uncertainty.

Maximum entropy (MaxEnt) principle has emerged as a prominent technique in the first category. One way to achieve MaxEnt is by minimizing the KL divergence between a uniform distribution and a target distribution. This is because a uniform distribution can guarantee full coverage of the space, which also displays maximum entropy. Recently, (Hazan et al., 2019) introduced the concept of maximum state entropy exploration (MSEE) in a broader spectrum of RL environments. Subsequently, multiple approaches have been proposed to enhance it, such as (Zhang et al., 2021; Seo et al., 2021; Yuan et al., 2022; Nedergaard & Cook, 2022; Yarats et al., 2021; Tiapkin et al., 2023), just to name a few. However, the utilization of multiple policies in these MaxEnt-based methods and the objective of uniform distribution over state space may cause the agent to spend a considerable amount of time near the starting states, leading to longer training time.

Conversely, *curiosity*-driven methods encourage exploration of unpredictable parts of the environment and prioritize the discovery of novel states beyond the initial ones. Usually, the dynamic of the environment is characterized by the transition probability of next state given current state-action pairs, i.e., $\mathbb{P}(\mathbf{s}_{t+1}|\mathbf{s}_t, \mathbf{a}_t)$. Hence, most approaches in this category use an auxiliary predictive model

$\mathbb{P}_\theta(\mathbf{s}_{t+1}|\mathbf{s}_t, \mathbf{a}_t)$ with parameters $\theta$, such as linear regression (Schmidhuber, 1991), convolution neural networks (Pathak et al., 2017) and fully-connected neural networks (Stadie et al., 2015; Yu et al., 2020; Pathak et al., 2017), to model the transition probability. Once the model is trained, intrinsic rewards can be defined with either the prediction error of the next state $\mathbf{s}_{t+1}$ or the information gain (Lopes & Mengue, 2022), which can be approximated by the difference between the estimate of the transition probability before and after new triple samples $\{\mathbf{s}_{t+1}, \mathbf{s}_t, \mathbf{a}_t\}$ are included.

The above-mentioned *curiosity*-driven techniques are model-based in the sense that they never explicitly estimate the true divergence of transition probability $\mathbb{P}(\mathbf{s}_{t+1}|\mathbf{s}_t, \mathbf{a}_t)$ from observations $\{\mathbf{s}_{t+1}, \mathbf{s}_t, \mathbf{a}_t\}_{t=1}^\infty$ in the trajectory. Rather, they model it implicitly with an internal parametric auxiliary model $\mathbb{P}_\theta(\mathbf{s}_{t+1}|\mathbf{s}_t, \mathbf{a}_t)$ for the ease of estimation. Hence, the exploration depends heavily on the predictive performance of the auxiliary models; and it is also hard for practitioners to decide which models to choose. If the model is well trained such that it learns precisely the conditional distribution $\mathbb{P}(\mathbf{s}_{t+1}|\mathbf{s}_t, \mathbf{a}_t)$, the RL agent may encounter vanishing intrinsic rewards. On the contrary, intrinsic rewards explode. Besides, the inclusion of auxiliary models introduces additional hyperparameters and parameters, making it challenging to maintain a balance between the model and the RL agent.

In this paper, we develop Maximum Conditional Divergence (MaxCondDiv), a new *curiosity*-driven exploration approach for exploration RL that does not rely on external rewards or parameterized models for prediction. To this end, akin to the MaxEnt principle, it leverages an information-theoretic measure to explicitly model and estimate the divergence of $\mathbb{P}(\mathbf{s}_{t+1}|\mathbf{s}_t, \mathbf{a}_t)$ in two adjacent trajectory fractions (i.e., $\max D(\mathbb{P}_c(\mathbf{s}_{t+1}|\mathbf{s}_t, \mathbf{a}_t); \mathbb{P}_f(\mathbf{s}_{t+1}|\mathbf{s}_t, \mathbf{a}_t))$, where c stands for "current" and f stands for "former") only based on observation triplets $\{\mathbf{s}_{t+1}, \mathbf{s}_t, \mathbf{a}_t\}$, in a model-free manner. Despite the simplicity of this idea, a precise estimation of the divergence between two conditional distributions is a non-trivial task, especially when the estimator is required to operate in a multivariate continuous space. To address this issue, we estimate $D(\mathbb{P}_c(\mathbf{s}_{t+1}|\mathbf{s}_t, \mathbf{a}_t); \mathbb{P}_f(\mathbf{s}_{t+1}|\mathbf{s}_t, \mathbf{a}_t))$ by introducing the notion of the Cauchy-Schwarz (CS) divergence (Principe et al., 2000; Yu et al., 2023), which significantly reduces the difficulty of estimation and enjoys several desirable properties than conventional Kullback–Leibler (KL) divergence and maximum mean discrepancy (MMD) (Gretton et al., 2012).

To summarize, we make the following key contributions:

- By explicitly modeling the conditional distribution $\mathbb{P}(\mathbf{s}_{t+1}|\mathbf{s}_t, \mathbf{a}_t)$ without training an auxiliary model, we propose MaxCondDiv as a new reward-free exploration strategy applicable to multivariate observations, which encourages the agent to explore divergent transition probability and leads to high information gain.

- Distinct to $f$-divergence such as KL divergence and integral probability metric such as MMD, the use of CS divergence simplifies the estimation and avoids unstable training.

- We also establish the connection between MaxCondDiv with respect to MaxEnt.

- Using MaxCondDiv, our agent can acquire intricate skills such as jumping and flipping in a fully reward-free environment. Using the visited states as a metric, our method outperforms other state-of-the-art reward-free exploration methods in three Mujoco environments.

## 2 BACKGROUND KNOWLEDGE AND RELATED WORKS

### 2.1 RÉNYI'S $\alpha$-ENTROPY AND CAUCHY-SCHWARZ DIVERGENCE

In information theory, a natural extension of the well-known Shannon's entropy is Rényi's $\alpha$-entropy (Rényi, 1961). For a random variable $\mathbf{x}$ with probability density function (PDF) $p(\mathbf{x})$ in a finite set $\mathcal{X}$, the $\alpha$-entropy $H_\alpha(\mathbf{x})$ is defined as:

$$H_\alpha(\mathbf{x}) = \frac{1}{1-\alpha} \log \int_\mathcal{X} p^\alpha(\mathbf{x}) d\mathbf{x}. \tag{1}$$

Similarly, for two random variables $\mathbf{x}$ and $\mathbf{y}$ with joint PDF $p(\mathbf{x}, \mathbf{y})$, the joint entropy is given by:

$$H_\alpha(\mathbf{x}, \mathbf{y}) = \frac{1}{1-\alpha} \log \int_\mathcal{Y} \int_\mathcal{X} p^\alpha(\mathbf{x}, \mathbf{y}) dx dy. \tag{2}$$

Thus, the $\alpha$-order mutual information[1] can be expressed as (Cachin, 1997; Teixeira et al., 2012):

$$I_\alpha(\mathbf{x}, \mathbf{y}) = H_\alpha(\mathbf{x}) + H_\alpha(\mathbf{y}) - H_\alpha(\mathbf{x}, \mathbf{y}). \tag{3}$$

Likewise, extensions for the relative entropy also exist; a modified version of Rényi's $\alpha$-relative entropy (or divergence) between PDFs $p$ and $q$ is given by (Lutwak et al., 2005):

$$D_\alpha(p; q) = \log \frac{(\int q^{\alpha-1} p)^{\frac{1}{1-\alpha}} (\int q^\alpha)^{\frac{1}{\alpha}}}{(\int p^\alpha)^{\frac{1}{\alpha(1-\alpha)}}}. \tag{4}$$

The limiting case of (1) and (4) for $\alpha \to 1$ are Shannon's entropy and KL divergence, respectively.

It turns out that for the case of $\alpha = 2$, the above quantities can be expressed as functions of inner products between PDFs, which makes them be easy to estimate in reproducing kernel Hilbert spaces (RKHS) (Principe, 2010). In particular, the quadratic entropy and divergence are given by:

$$H_2(\mathbf{x}) = -\log \int_\mathcal{X} p^2(\mathbf{x}) dx, \quad \text{and} \quad D_{\text{CS}}(p; q) = -\frac{1}{2} \log \frac{(\int pq)^2}{(\int p^2)(\int q^2)}. \tag{5}$$

Eq. (5) is also called the Cauchy-Schwarz (CS) divergence as it can be obtained by applying the CS inequality associated with $p(\mathbf{x})$ and $q(\mathbf{x})$:

$$\left| \int p(\mathbf{x}) q(\mathbf{x}) d\mathbf{x} \right|^2 \leq \int |p(\mathbf{x})|^2 \, d\mathbf{x} \int |q(\mathbf{x})|^2 \, d\mathbf{x}. \tag{6}$$

The CS inequality also holds for two conditional distributions $p(\mathbf{y}|\mathbf{x})$ and $q(\mathbf{y}|\mathbf{x})$ (Yu et al., 2023), the resulting conditional CS divergence can be expressed naturally as:

$$D_{\text{CS}}(p(\mathbf{y}|\mathbf{x}); q(\mathbf{y}|\mathbf{x}))$$

$$= -2\log \left( \int_\mathcal{X} \int_\mathcal{Y} p(\mathbf{y}|\mathbf{x}) q(\mathbf{y}|\mathbf{x}) d\mathbf{x} d\mathbf{y} \right) + \log \left( \int_\mathcal{X} \int_\mathcal{Y} p^2(\mathbf{y}|\mathbf{x}) d\mathbf{x} d\mathbf{y} \right) + \log \left( \int_\mathcal{X} \int_\mathcal{Y} q^2(\mathbf{y}|\mathbf{x}) d\mathbf{x} d\mathbf{y} \right)$$

$$= -2\log \left( \int_\mathcal{X} \int_\mathcal{Y} \frac{p(\mathbf{x}, \mathbf{y}) q(\mathbf{x}, \mathbf{y})}{p(\mathbf{x}) q(\mathbf{x})} d\mathbf{x} d\mathbf{y} \right) + \log \left( \int_\mathcal{X} \int_\mathcal{Y} \frac{p^2(\mathbf{x}, \mathbf{y})}{p^2(\mathbf{x})} d\mathbf{x} d\mathbf{y} \right) + \log \left( \int_\mathcal{X} \int_\mathcal{Y} \frac{q^2(\mathbf{x}, \mathbf{y})}{q^2(\mathbf{x})} d\mathbf{x} d\mathbf{y} \right).$$

$$\tag{7}$$

## 2.2 Intrinsically-Motivated Exploration RL

The existing intrinsically motivated exploration approaches can be broadly categorized into two types: the *space coverage* and the *curiosity*-driven approaches. Approaches rooted in the MaxEnt principle for achieving *space coverage* have gained popularity recently due to their strong mathematical interpretability and performance. For example, maximum state entropy exploration (MSEE) by (Hazan et al., 2019) guarantees uniform coverage of the state space. It offers a proof of policy improvement when utilizing the APPROXPLAN/DENSITYEST oracle. Later, MaxRényi (Zhang et al., 2021) replaced the Shannon entropy with Rényi entropy, and maximizes the entropy in the joint space of action and state. RE3 (Seo et al., 2021) incorporates neural encoders, enabling their application in video-oriented environments like Atari. RISE (Yuan et al., 2022) integrates both RE3 and MaxRényi, leveraging them to accelerate the learning process. In (Nedergaard & Cook, 2022) and (Yarats et al., 2021), $k$-means and prototypical representations are introduced to enhance the quality of latent vectors. (Tiapkin et al., 2023) studies MSEE to learn a policy leading to $\epsilon$-optimal maximum and reduces the sample complexity.

The other type, *curiosity*-driven approaches, have their roots traced back to the 70's when the concept of "observer's information" and "interestingness" were introduced (Pfaffelhuber, 1972; Lenat, 1976). Recent popular prediction error-based approaches, largely driven by advancements in deep neural networks (DNNs), fall under this category. For instance, ICM (Pathak et al., 2017) utilizes CNN as the auxiliary model to predict the next image, whereas GIRIL(Yu et al., 2020) implements variational

---

[1]There is no generally accepted definition on $\alpha$-order mutual information (Verdú, 2015). We took the one (Cachin, 1997) that is inspired by the strong chain rule of Shannon entropy, i.e., $H(\mathbf{x}, \mathbf{y}) = H(\mathbf{x}) + H(\mathbf{y}|\mathbf{x})$.

autoencoder (VAE) (Kingma & Welling, 2014) to model the transitions in environments. Similarly, (Shyam et al., 2019) aims to maximize the Jensen-Shannon divergence of fully-connected neural network outputs. In contrast to these methods, we pursue a model-free approach.

Our method shares the closest resemblance with the model-free *curiosity*-driven approach introduced by (Storck et al., 1995), which estimates the transition probability directly from observations. However, this method can only be used in tabular discrete environments, as it calculates the transition probability by counting. In contrast, our method is applicable to both discrete and continuous environments and is compatible with arbitrary RL techniques, thanks to the use of CS divergence.

# 3 MAXIMUM CONDITIONAL DIVERGENCE (MAXCONDDIV) EXPLORATION

The intrinsically-Motivated exploration RL problem can be defined as policy search in an infinite-horizon Markov decision process (MDP) defined by a 6-tuple $(\mathcal{S}, \mathcal{A}, p_{\mathbf{s}}, r^{\mathrm{E}}, r^{\mathrm{I}}, \gamma)$, where $\mathcal{S}$ is the set of all possible states, $\mathcal{A}$ is the set of all possible actions. $p_{\mathbf{s}}(\mathbf{s}_{t+1}|\mathbf{s}_t, \mathbf{a}_t)$ is the transition probability density of the next state $\mathbf{s}_{t+1} \in \mathcal{S}$ given the current state $\mathbf{s}_t \in \mathcal{S}$ and action $\mathbf{a}_t \in \mathcal{A}$. The environment omits extrinsic rewards given by the extrinsic reward function $r^{\mathrm{E}}(\mathbf{s}_t, \mathbf{a}_t)$. Meanwhile, the intrinsic reward function $r^{\mathrm{I}}(\rho_{t-})$ determines the intrinsic rewards based on historical data $\rho_{t-}$ collected before time step $t$. $\gamma \in [0, 1)$ is a discount factor. The optimal policy aims to learn a policy $\pi(\mathbf{a}_t|\mathbf{s}_t) : \mathcal{S} \mapsto \mathcal{A}$ by maximizing extrinsic and intrinsic rewards:

$$\pi^* = \operatorname{argmax}_\pi \mathbb{E}_{\rho \sim \pi} \left( \sum_{t=0}^{T-1} \gamma^t [r^{\mathrm{E}}(\mathbf{s}_t, \mathbf{a}_t) + \beta r^{\mathrm{I}}(\rho_{t-})] \right), \tag{8}$$

where $\beta$ is a hyperparameter that determines the relative importance of intrinsic and extrinsic reward, and $\rho = \{\mathbf{s}_{t+1}, \mathbf{s}_t, \mathbf{a}_t\}_{t=0}^{T-1}$ is the data collection by executing policy $\pi$. We specifically consider the case of reward-free cases, where the extrinsic reward $r^{\mathrm{E}}(\mathbf{s}_t, \mathbf{a}_t)$ is consistently zero. Our method aims to design an intrinsic reward function $r^{\mathrm{I}}(\rho_{t-})$ for exploring the functional space of transitions $p_{\mathbf{s}}(\mathbf{s}_{t+1}|\mathbf{s}_t, \mathbf{a}_t)$, without relying on any extrinsic reward $r^{\mathrm{E}}(\mathbf{s}_t, \mathbf{a}_t)$.

## 3.1 CONDITIONAL CAUCHY-SCHWARZ DIVERGENCE (CCSD) REWARD FUNCTION

Our focus is on enabling the agent to acquire novel transitions in contrast to recent visited samples. To specify a transition sample, we require a triplet sample consisting of the next state $\mathbf{s}_{t+1}$, the current state $\mathbf{s}_t$, and the action $\mathbf{a}_t$. Hence, a complete trajectory $\mathcal{T}_E = \{(\mathbf{s}_2, \mathbf{s}_1, \mathbf{a}_1), (\mathbf{s}_3, \mathbf{s}_2, \mathbf{a}_2), \cdots, (\mathbf{s}_T, \mathbf{s}_{T-1}, \mathbf{a}_{T-1}), \cdots\}$, $E$ for "entire", is defined to be the sequence of triplet samples, as illustrated in Fig. 1. Meanwhile, we utilize a first-in-first-out replay buffer (sliding window) to locally store transition samples. We refer to this subsequence of trajectory $\mathcal{T}_E$ as a "trajectory fraction", denoted as $\mathcal{T}$. The trajectory fraction $\mathcal{T}$ contains data for a maximum of $2\tau$ previous time steps. Then we define $\mathbb{P}_{\mathrm{f}}(\mathbf{s}_{t+1}|\mathbf{s}_t, \mathbf{a}_t) = \mathbb{P}_{(\mathbf{s}_{t+1}, \{\mathbf{s}_t, \mathbf{a}_t\}) \sim \mathcal{T}_{1:\tau}}(\mathbf{s}_{t+1}|\mathbf{s}_t, \mathbf{a}_t)$, f for "former", as the transition probability $\mathbb{P}(\mathbf{s}_{t+1}|\mathbf{s}_t, \mathbf{a}_t)$ for triplets that are sampled from $\mathcal{T}_{1:\tau}$ (i.e., 1-st to $\tau$-th elements of $\mathcal{T}$). Similarly, $\mathbb{P}_{\mathrm{c}}(\mathbf{s}_{t+1}|\mathbf{s}_t, \mathbf{a}_t) = \mathbb{P}_{(\mathbf{s}_{t+1}, \{\mathbf{s}_t, \mathbf{a}_t\}) \sim \mathcal{T}_{(\tau+1):2\tau}}(\mathbf{s}_{t+1}|\mathbf{s}_t, \mathbf{a}_t)$, c for "current", be the transition probability $\mathbb{P}(\mathbf{s}_{t+1}|\mathbf{s}_t, \mathbf{a}_t)$ for triplets that are sampled from $\mathcal{T}_{\tau+1:2\tau}$. Our framework estimates an intrinsic reward defined as the divergence between $\mathbb{P}_{\mathrm{f}}(\mathbf{s}_{t+1}|\mathbf{s}_t, \mathbf{a}_t)$ and $\mathbb{P}_{\mathrm{c}}(\mathbf{s}_{t+1}|\mathbf{s}_t, \mathbf{a}_t)$, and learns a policy by maximizing this conditional divergence (MaxCondDiv). More formally, our optimal policy aims to maximize the divergence between "former" and "current" transitions in the trajectory fraction $\mathcal{T}$:

$$\pi^*_{\mathrm{MaxCondDiv}} = \operatorname{argmax}_\pi \mathbb{E}_{\rho \sim \pi} \left( \sum_{\mathcal{T} \in [\mathcal{T}_E]} D(\mathbb{P}_{\mathrm{c}}(\mathbf{s}_{t+1}|\mathbf{s}_t, \mathbf{a}_t); \mathbb{P}_{\mathrm{f}}(\mathbf{s}_{t+1}|\mathbf{s}_t, \mathbf{a}_t))) \right). \tag{9}$$

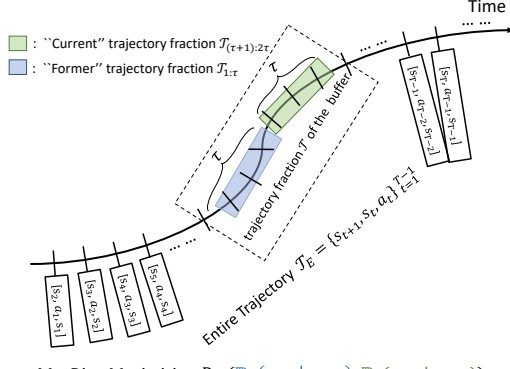

MaxDiv: Maximizing $D_{cs}(\mathbb{P}_f(s_{t+1}|s_t, a_t); \mathbb{P}_c(s_{t+1}|s_t, a_t))$

**Figure 1:** The structure of our replay buffer. We choose the length of $\mathcal{T}$ to be $2\tau$ and divide it equally into "current" and "former" parts. The split point is arbitrary, and overlapping fractions are also possible. If we designate the $\mathcal{T}_{1:2\tau-1}$ as "former" and the $\mathcal{T}_{1:2\tau}$ samples as "current", our approach is consistent with that of (Storck et al., 1995).

**Algorithm 1:** MaxCondDiv Exploration Reinforcement Learning

---

1: Initialize data collections $\rho$ ; empty buffer $\mathcal{T}$.
2: **while** not converge **do**
3:     clear $\mathcal{T}$.
4:     **while** the episode not end **do**
5:         Sample the action $a_t \sim \pi(a_t|s_t)$.
6:         Sample $s_{t+1} \sim P(s_{t+1}|s_t, a_t)$.
7:         Record trio $\{s_{t+1}, s_t, a_t\}$ into the buffer $\mathcal{T}$.
8:         Compute conditional divergence $\widehat{D}_{\mathrm{CS}}(\mathbb{P}_{\mathrm{f}}; \mathbb{P}_{\mathrm{c}})$ with Eq. (13) from triplets in the buffer $\mathcal{T}$.
9:         Record $\{s_{t+1}, s_t, a_t, (\widehat{D}_{\mathrm{CS}})_t\}$ in the data collection $\rho$
10:        update policy via sampling from $\rho$.
11:    **end while**
12: **end while**

---

### 3.2 WHY CONDITIONAL CAUCHY-SCHWARZ DIVERGENCE (CCSD) FOR MAXCONDDIV?

In principle, any divergence could be used in the context of MaxCondDiv. We underscore the rationale for choosing CS divergence, rather than the popular KL divergence and MMD. For KL divergence $D_{\mathrm{KL}}(p; q) = \int p\log(\frac{p}{q})$, its conditional extension follows a decomposition rule (Cover, 1999):

$$
\begin{aligned}
D_{\mathrm{KL}}(\mathbb{P}_{\mathrm{c}}(\mathbf{s}_{t+1}|\mathbf{s}_t, \mathbf{a}_t); \mathbb{P}_{\mathrm{f}}(\mathbf{s}_{t+1}|\mathbf{s}_t, \mathbf{a}_t)) = {} & D_{\mathrm{KL}}(\mathbb{P}_{\mathrm{c}}(\mathbf{s}_{t+1}, \mathbf{s}_t, \mathbf{a}_t); \mathbb{P}_{\mathrm{f}}(\mathbf{s}_{t+1}, \mathbf{s}_t, \mathbf{a}_t)) \\
& - D_{\mathrm{KL}}(\mathbb{P}_{\mathrm{c}}(\mathbf{s}_t, \mathbf{a}_t); \mathbb{P}_{\mathrm{f}}(\mathbf{s}_t, \mathbf{a}_t)),
\end{aligned}
\tag{10}
$$

in which both terms are usually evaluated by $k$-NN estimator (Wang et al., 2009). However, the term $\log(\frac{p}{q})$ will explode when $q \to 0$, a scenario commonly encountered in our RL experiments. This instability can disrupt the learning process for RL agents. Further empirical details regarding MaxCondDiv's use of KL divergence can be found in both Section 4.3 and the Appendix C.2.

Fortunately, CS divergence does not have this issue: it is much more stable and never explodes (see also our discussions in Appendix D.1). Theoretically, CS divergence is no greater than KL divergence in Gaussian distributed data. Therefore, it provides a viable alternative objective when KL divergence is hard to be applied in practice.

**Proposition 1.** *For two arbitrary d-variate Gaussian distributions* $p \sim \mathcal{N}(\mu_1, \Sigma_1)$ *and* $q \sim \mathcal{N}(\mu_2, \Sigma_2)$, *we have:*

$$
D_{CS}(p; q) \leq \min\{D_{KL}(p; q), D_{KL}(q; p)\}.
\tag{11}
$$

All Proofs can be found in Appendix A. Moreover, compared with the $k$-NN estimator, our empirical estimator of CS divergence is differentiable, which makes it promising for potential applications in deep multi-modal learning, where the RL module may play a critical role.

MMD embeds probability functions in a reproducing kernel Hilbert space (RKHS). If we take the conditional MMD definition in (Ren et al., 2016), the estimator involves matrix inverse and an extra hyper-parameter, which also makes the training highly unstable and time consuming. See the Appendix C.1 for more discussions.

In this paper, we suggest conditional Cauchy-Schwarz divergence (CCSD) for MaxCondDiv:

$$D_{\mathrm{CS}}(\mathbb{P}_{\mathrm{f}}(\mathbf{s}_{t+1}|\mathbf{s}_t, \mathbf{a}_t); \mathbb{P}_{\mathrm{c}}(\mathbf{s}_{t+1}|\mathbf{s}_t, \mathbf{a}_t))$$

$$= -2\log\left(\int_{\mathcal{S}_{t+1}}\int_{\{\mathcal{S}_t, \mathcal{A}_t\}} \frac{\mathbb{P}_{\mathrm{f}}(\mathbf{s}_{t+1}, \{\mathbf{s}_t, \mathbf{a}_t\})\mathbb{P}_{\mathrm{c}}(\mathbf{s}_{t+1}, \{\mathbf{s}_t, \mathbf{a}_t\})}{\mathbb{P}_{\mathrm{f}}(\{\mathbf{s}_t, \mathbf{a}_t\})\mathbb{P}_{\mathrm{c}}(\{\mathbf{s}_t, \mathbf{a}_t\})} d\{\mathbf{s}_t, \mathbf{a}_t\}d\mathbf{s}_{t+1}\right)$$

$$+ \log\left(\int_{\mathcal{S}_{t+1}}\int_{\{\mathcal{S}_t, \mathcal{A}_t\}} \frac{\mathbb{P}_{\mathrm{f}}^2(\mathbf{s}_{t+1}, \{\mathbf{s}_t, \mathbf{a}_t\})}{\mathbb{P}_{\mathrm{f}}^2(\{\mathbf{s}_t, \mathbf{a}_t\})} d\{\mathbf{s}_t, \mathbf{a}_t\}d\mathbf{s}_{t+1}\right) \tag{12}$$

$$+ \log\left(\int_{\mathcal{S}_{t+1}}\int_{\{\mathcal{S}_t, \mathcal{A}_t\}} \frac{\mathbb{P}_{\mathrm{c}}^2(\mathbf{s}_{t+1}, \{\mathbf{s}_t, \mathbf{a}_t\})}{\mathbb{P}_{\mathrm{c}}^2(\{\mathbf{s}_t, \mathbf{a}_t\})} d\{\mathbf{s}_t, \mathbf{a}_t\}d\mathbf{s}_{t+1}\right),$$

where $\mathcal{S}_{t+1}, \{\mathcal{S}_t, \mathcal{A}_t\}$ are world set of $\mathbf{s}_{t+1}$ and $\{\mathbf{s}_t, \mathbf{a}_t\}$, respectively.

### 3.3 PRACTICAL METHODS FOR ACCURATELY ESTIMATING THE CCSD INTRINSIC REWARD

**Proposition 2** (Empirical Estimator of $D_{\mathrm{CS}}(\mathbb{P}_{\mathrm{f}}(\mathbf{s}_{t+1}|\mathbf{s}_t, \mathbf{a}_t); \mathbb{P}_{\mathrm{c}}(\mathbf{s}_{t+1}|\mathbf{s}_t, \mathbf{a}_t))$ (Yu et al., 2023))**. *Given observations in the $2\tau$-length trajectory fraction $\mathcal{T} = \{[(\mathbf{s}_{t+1})_i, \{\mathbf{s}_t, \mathbf{a}_t\}_i]\}_{i=1}^{2\tau}$, dividing them into two fractions such that $\{[(\mathbf{s}_{t+1})_i, \{\mathbf{s}_t, \mathbf{a}_t\}_i]\}_{i=1}^{\tau}$ are sampled from distribution $\mathbb{P}_{\mathrm{f}}(\mathbf{s}_{t+1}, \{\mathbf{s}_t, \mathbf{a}_t\})$ and $\{[(\mathbf{s}_{t+1})_i, \{\mathbf{s}_t, \mathbf{a}_t\}_i]\}_{i=\tau+1}^{2\tau}$ are sampled from $\mathbb{P}_{\mathrm{c}}(\mathbf{s}_{t+1}, \{\mathbf{s}_t, \mathbf{a}_t\})$. Let $K^{\mathrm{f}}$ and $L^{\mathrm{f}}$ denote, respectively, the Gram matrices[2] for the concatenated variable $\{\mathbf{s}_t, \mathbf{a}_t\}$ and the variable $\mathbf{s}_{t+1}$ in the distribution $\mathbb{P}_{\mathrm{f}}$. That is, $(K^{\mathrm{f}})_{ij} = \kappa(\{\mathbf{s}_t, \mathbf{a}_t\}_i - \{\mathbf{s}_t, \mathbf{a}_t\}_j)$, $(L^{\mathrm{f}})_{ij} = \kappa(\{\mathbf{s}_{t+1}\}_i - \{\mathbf{s}_{t+1}\}_j)$ for $i, j = 1 : \tau$, in which $\kappa$ is a Gaussian kernel and takes the form of $\kappa = \exp\left(-\frac{\|\cdot\|^2}{2\sigma^2}\right)$. Similarly, let $K^{\mathrm{c}}$ and $L^{\mathrm{c}}$ denote, respectively, the Gram matrices for the variable $\{\mathbf{s}_t, \mathbf{a}_t\}$ and the variable $\mathbf{s}_{t+1}$ in the distribution $\mathbb{P}_{\mathrm{c}}$. Meanwhile, let $K^{\mathrm{fc}} \in \mathbb{R}^{\tau \times \tau}$ (i.e., $(K^{\mathrm{fc}})_{ij} = \kappa(\{\mathbf{s}_t, \mathbf{a}_t\}_i - \{\mathbf{s}_t, \mathbf{a}_t\}_j)$, $i = 1 : \tau$ and $j = \tau + 1 : 2\tau$) denote the Gram matrix for variable $\{\mathbf{s}_t, \mathbf{a}_t\}$ from distribution $\mathbb{P}_{\mathrm{f}}$ to distribution $\mathbb{P}_{\mathrm{c}}$, and $L^{\mathrm{fc}} \in \mathbb{R}^{\tau \times \tau}$ the Gram matrix for variable $\mathbf{s}_{t+1}$ from distribution $\mathbb{P}_{\mathrm{f}}$ to distribution $\mathbb{P}_{\mathrm{c}}$. The Gram matrices $K^{\mathrm{cf}}$ and $L^{\mathrm{cf}}$ can be defined similarly. The empirical estimation of $D_{CS}(\mathbb{P}_{\mathrm{f}}(\mathbf{s}_{t+1}|\mathbf{s}_t, \mathbf{a}_t); \mathbb{P}_{\mathrm{c}}(\mathbf{s}_{t+1}|\mathbf{s}_t, \mathbf{a}_t))$ is given by:*

$$\widehat{D}_{CS}(\mathbb{P}_{\mathrm{f}}(\mathbf{s}_{t+1}|\mathbf{s}_t, \mathbf{a}_t); \mathbb{P}_{\mathrm{c}}(\mathbf{s}_{t+1}|\mathbf{s}_t, \mathbf{a}_t)) = \log\left(\sum_{j=1}^{\tau}\left(\frac{\sum_{i=1}^{\tau} K_{ji}^{\mathrm{f}} L_{ji}^{\mathrm{f}}}{(\sum_{i=1}^{\tau} K_{ji}^{\mathrm{f}})^2}\right)\right) + \log\left(\sum_{j=1}^{\tau}\left(\frac{\sum_{i=1}^{\tau} K_{ji}^{\mathrm{c}} L_{ji}^{\mathrm{c}}}{(\sum_{i=1}^{\tau} K_{ji}^{\mathrm{c}})^2}\right)\right)$$

$$- \log\left(\sum_{j=1}^{\tau}\left(\frac{\sum_{i=1}^{\tau} K_{ji}^{\mathrm{fc}} L_{ji}^{\mathrm{fc}}}{(\sum_{i=1}^{\tau} K_{ji}^{\mathrm{c}})(\sum_{i=1}^{\tau} K_{ji}^{\mathrm{fc}})}\right)\right) - \log\left(\sum_{j=1}^{\tau}\left(\frac{\sum_{i=1}^{\tau} K_{ji}^{\mathrm{cf}} L_{ji}^{\mathrm{cf}}}{(\sum_{i=1}^{\tau} K_{ji}^{\mathrm{cf}})(\sum_{i=1}^{\tau} K_{\mathrm{cf}}^{\mathrm{f}})}\right)\right). \tag{13}$$

We offer a visualization in Appendix D.2 and provide implementation in Appendix E to facilitate comprehension of the Gram matrix. The estimator exhibits low computational complexity with $\mathcal{O}(N^2)$. The CCSD intrinsic reward can be combined with any RL methods, e.g., Q-learning (Watkins & Dayan, 1992), PPO (Schulman et al., 2017). We summarize the training pseudo code in Algorithm 1.

### 3.4 CONNECTION BETWEEN MAXCONDDIV AND MAXENT

**Proposition 3.** *Let $X$ and $Y$ be two random variable with marginal PDFs $p_X(x) = p_X(X = x)$ and $p_Y(y) = p_Y(Y = y)$, respectively, where $x \in \mathcal{R}$ and $y \in \mathcal{R}$. Let $p_{XY}(x, y) = p_{XY}(X = x, Y = y)$ denotes the joint PDF. We have:*

$$\frac{1}{2}H_2(x) + \frac{1}{2}H_2(y) - I_2(x, y) \geq D_{cs}(p_X; p_Y) \tag{14}$$

*iff:*

$$\int_{r \in \mathcal{R}} p_X(X = r)p_Y(Y = r)dr \geq \int_{y \in \mathcal{R}}\int_{x \in \mathcal{R}} p_{XY}^2(x, y)dxdy \tag{15}$$

---

[2]In kernel learning, the Gram or kernel matrix is a symmetric matrix where each entry is the inner product of the corresponding data points in a reproducing kernel Hilbert space (RKHS), defined by kernel function $\kappa$.

*where $H_2(\cdot)$ and $I_2(\cdot, \cdot)$ are 2nd-order Rényi entropy and mutual information, as defined in Eq.( 1) and Eq. (3), respectively.*

We justify Proposition 3 in the Appendix A.3. For two variables $X$ and $Y$, maximizing their CS divergence $D_{cs}(p_X; p_Y)$ also maximizes a lower bound of the sum of 2nd-order Rényi entropy of $X$ and $Y$ minus their 2nd-order Rényi mutual information. It applies to our case by substituting $X$ and $Y$ with $\mathbf{s}_{t+1} \sim \mathbb{P}_f(\cdot|\mathbf{s}_t, \mathbf{a}_t)$ and $\mathbf{s}_{t+1} \sim \mathbb{P}_c(\cdot|\mathbf{s}_t, \mathbf{a}_t)$, respectively. Therefore, maximizing our CCSD is closely related to the maximum trajectory entropy exploration (Ekroot & Cover, 1993; Fiechter, 1994), i.e., $\text{argmax}_\pi H_{traj}(p_{traj}^\pi)$, where $p_{traj}^\pi = \pi(\mathbf{a}_1|\mathbf{s}_1) \prod_{t=2}^{2\tau} \mathbb{P}_{t-1}(\mathbf{s}_t|\mathbf{s}_{t-1}, \mathbf{a}_{t-1}) \pi_t(\mathbf{a}_t|\mathbf{s}_t)$, which can also be obtained by solving entropy regularized Bellman equations using entropy of the transition probabilities $H(\mathbf{s}_{t+1}|\mathbf{s}_t, \mathbf{a}_t)$ as rewards (Fiechter, 1994; Tiapkin et al., 2023). Meanwhile, the last term in Eq. (15) incentives the independence between "former" and "current".

# 4 EXPERIMENTS

## 4.1 A THOUGHT EXPERIMENT

To highlight the contrast between MaxEnt and MaxCondDiv, let us consider a thought experiment visualized in Fig. 2, where optimal policies are realized through a retrospective step: The scenario involves a 2-D open-world environment. In each trial, the agent initiates from the central position $(100, 100)$ and undergoes a sequence of 200 time steps. At each time step, the agent's action is to select an arbitrary direction and move one unit distance. We replicate this procedure 100 times for each exploration policy.

As illustrated in Fig. 2, the random policy keeps the agent predominantly near its starting point. Conversely, the optimal MaxEnt policy distributes the agent's trajectory evenly, covering a range that far exceeds what a random policy achieves. Our MaxCondDiv agent selects a random direction to move in the first step because the sample in the buffer is $(100, 100)$, and moving in any radial direction is equally divergent. For instance, if the agent moves to $(100, 101)$, the samples in the buffer become $[(100, 100), (100, 101)]$. To maximize the divergence, the agent needs to move to $(100, 102)$ in the next step. Deviating from this will result in smaller distances to the previously visited states. Consequently, during each trial, the agent consistently moves in one single random direction. Over 100 trials, the MaxCondDiv agent explores the world more radially, akin to a fireworks display. We also explore maximizing the divergence between joint distributions (MaxJDiv), i.e., $D_{\text{CS}}(\mathbb{P}_f(\mathbf{s}_{t+1}, \mathbf{s}_t, \mathbf{a}_t); \mathbb{P}_c(\mathbf{s}_{t+1}, \mathbf{s}_t, \mathbf{a}_t))$, an alternative to MaxCondDiv. For the joint probability $\mathbb{P}(\mathbf{s}_{t+1}, \mathbf{s}_t, \mathbf{a}_t) = \mathbb{P}(\mathbf{s}_{t+1}|\mathbf{s}_t, \mathbf{a}_t)\mathbb{P}(\mathbf{s}_t, \mathbf{a}_t)$, if $\mathbb{P}(\mathbf{s}_t, \mathbf{a}_t)$ is small (that is, the corresponding state-action pairs are not fully explored), the corresponding $\mathbb{P}(\mathbf{s}_{t+1}|\mathbf{s}_t, \mathbf{a}_t)$ plays a minor role in the learning objective. If $\mathbb{P}(\mathbf{s}_t, \mathbf{a}_t)$ is large, the corresponding $\mathbb{P}(\mathbf{s}_{t+1}|\mathbf{s}_t, \mathbf{a}_t)$ would have higher weight. This is in contrast to our goal, in which we expect that regions with low $\mathbb{P}(\mathbf{s}_t, \mathbf{a}_t)$ should be explored more during exploration. Hence, we expect the performance of MaxJDiv is outperformed by MaxCondDiv.

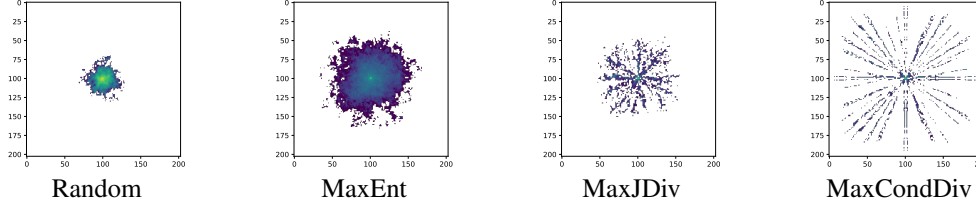

| Random | MaxEnt | MaxJDiv | MaxCondDiv |

**Figure 2:** The realization of the thought experiment using random, optimal MaxEnt, MaxJDiv and MaxCond-Div policies. The MaxEnt principle facilitates exploration by uniformly visiting more states, whereas our MaxCondDiv principle guides exploration by maintaining distance from previously visited states.

## 4.2 RESULTS ON MOUNTAINCAR AND MAZE

In this section, we experiment with MountainCar and Maze, using Q-learning as the oracle, and compare it to the MaxEnt principle and random policy. For MaxEnt principle, we adopt the MSEE of (Hazan et al., 2019). The agent is trained 100 episodes, i.e., around 50,000 steps. In Figure 3

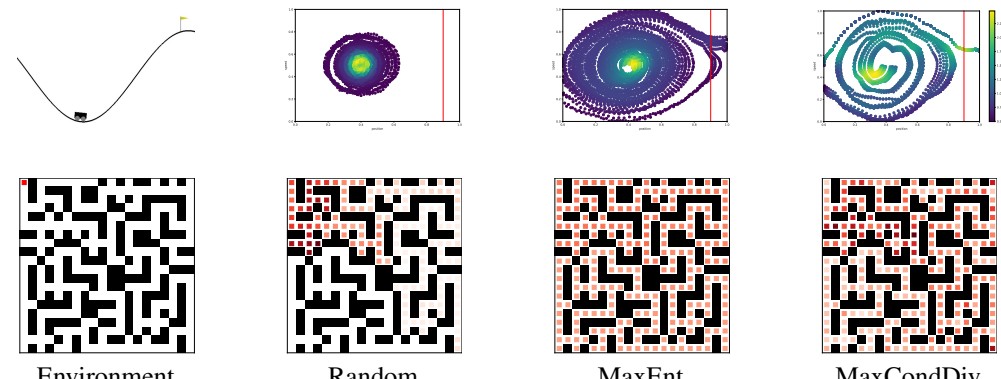

| Environment | Random | MaxEnt | MaxCondDiv |

**Figure 3:** Trajectories of different trained policies on Mountain Car and Maze. The flag positions are indicated by red vertical lines. Both MaxEnt and MaxCondDiv can facilitate environment exploration to achieve a defined goal. MaxEnt emphasizes uniform visits to all states, while our MaxCondDiv strategy involves maintaining distance from starting points.

(top-left), we illustrate the MountainCar environment, where the most challenging state to explore is indicated by the flag. We executed trained policies, and visualize their trajectories using kernel density estimation in heatmaps. As expected, the random policy fails to reach the flag and remains close to the starting points. Although MaxEnt can reach the flag, it focuses more on states near the starting point. In contrast, MaxCondDiv reaches the flag more frequently but tends to ignore regions near visited states.

In Maze, as shown in Fig. 3, the agent drives the red point to explore the maze. We record trajectories for $50,000$ steps. The agent is reset to the start point every $1,000$ steps. The random policy remains near the start points, while MaxEnt explores the entire state space evenly. Our MaxCondDiv also explores the entire maze, but tends to stay away from the start point. In the heatmap of MaxCondDiv, the probability at the start point is much lower than that at challenging states, such as top-right and bottom-right corner, indicating that our method has a higher probability to "reach the boundary".

## 4.3 RESULTS ON MUJOCO

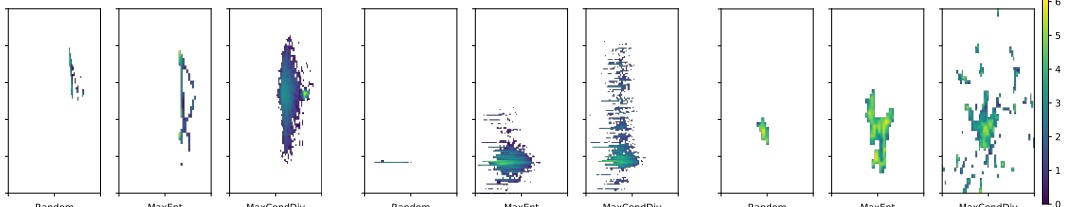

**Figure 4:** Trajectories of various trained policies on Mujoco. Consistent with prior findings, our MaxCondDiv approach is characterized by a deliberate maintenance of distance from visited states.

Mujoco is an advanced physics simulation with continuous spaces and multiple tasks in which we select Hopper, Halfcheetah and Ant. In our experiments, observation noise is introduced by rounding state and action values to two decimal places, and the RL backbone is a PPO agent. Details of hyper-parameters are in Appendix B. The agent is trained for 1,000 episodes, i.e., 1,000,000 steps in total. The agent restarts from the initial state with uniform noise in each episode.

**Divergence vs Entropy.** We depict the distribution of visited states within 10,000 steps using trained agents in Fig. 4. For both Hopper and Halfcheetah, we visualize the first two states, which are the z-coordinate of the front tip and the top angle for Hopper, and the x-y coordinates for Ant. In Hopper, our method outperforms others and effectively learns the necessary degrees of freedom to walk forward or backward. In HalfCheetah and Ant, MaxEnt explores a broad space, generating trajectories evenly distributed around the start point. Our MaxCondDiv diverges by concentrating

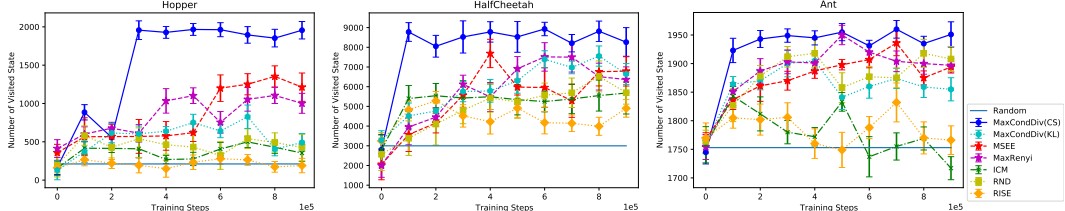

**Figure 5:** Learning curves tracked over 10 checkpoints, employing visited states acquired by executing trained policies for 10,000 steps as the Y-axis. Our MaxCondDiv method demonstrates superior performance in both exploration range and speed compared to the baseline methods.

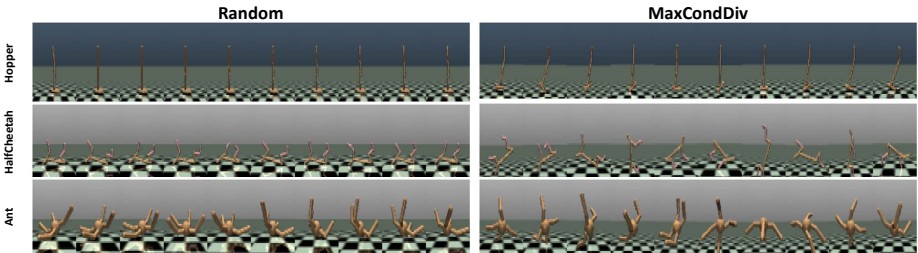

**Figure 6:** The policies trained with MaxCondDiv using no extrisic rewards exhibit diverse behaviors.

on exploration far from the start point, as confirmed by the radial exploration trajectories from our thought experiment.

**Comparsion to SOTA Exploration RL Approaches.** We compare our method with random policy, curiosity-driven exploration (Pathak et al., 2017)(ICM), Exploration by Random Network Distillation (Burda et al., 2019) (RND), Rényi State Entropy Maximization (Yuan et al., 2022) (RISE), Exploration by Maximizing Rényi Entropy (MaxRényi), Maximum State Entropy Exploration (Hazan et al., 2019) (MSEE) and our MaxCondDiv using KL divergene. Similar to (Hazan et al., 2019), we evaluate the exploration performance of trained agents using the number of visited states. It is widely accepted that a large number of visited states will result in improved performance in downstream tasks because the agent can gather sufficient information (Jin et al., 2020). We execute the trained policy for 10,000 steps every 100,000 steps of training, depicting the numbers of visited states in Fig. 5. MaxCondDiv outperforms the baseline methods in terms of exploration range and training speed. Furthermore, we carry out experiments on downstream tasks which is commonly referred to as the "planning phase" within a reward-free RL framework (Jin et al., 2020). However, the results in downstream tasks are significantly influenced by the subsequent offline RL algorithms employed and are limited compared to online RL. Consequently, we have not utilized them as our primary results but have included them in Appendix C.5 for reference.

**Learned Skill without Using Extrinsic Rewards.** We have included images depicting agent motions in Fig. 6. MaxCondDiv acquires a range of basic behaviors such as jumping forward, flipping, etc, without extrinsic rewards. Videos are shown in Appendix C.3 and attached zip file.

## 5 CONCLUSION

We propose Mamaximum Conditional Divergence (MaxCondDiv), a model-free method for exploration without extrinsic rewards that estimates the difference of transition probabilities in two trajectory fractions using a conditional Cauchy-Schwarz divergence estimator. MaxCondDiv exhibits distinct exploration behaviors compared to maximum entropy principle and avoids auxiliary model selection bias observed in other curiosity-driven approaches. We evaluate MaxCondDiv in two discrete and three continuous environments, consistently achieving exploration of more states or successfully reaching challenging states.

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

# A   Proofs of Main Results

## A.1   Proof of the Proposition 1

**Proposition 1.** *For two arbitrary d-variate Gaussian distributions $p \sim \mathcal{N}(\mu_1, \Sigma_1)$ and $q \sim \mathcal{N}(\mu_2, \Sigma_2)$,*

$$D_{CS}(p; q) \leq \min\{D_{KL}(p; q), D_{KL}(q; p)\}. \tag{16}$$

*Proof.* Given two $d$-dimensional Gaussian distributions $p \sim \mathcal{N}(\mu_1, \Sigma_1)$ and $q \sim \mathcal{N}(\mu_2, \Sigma_2)$, the KL divergence for $p$ and $q$ is given by:

$$D_{\text{KL}}(p; q) = \frac{1}{2}\left(\text{tr}(\Sigma_2^{-1}\Sigma_1) - d + (\mu_2 - \mu_1)^T\Sigma_2^{-1}(\mu_2 - \mu_1) + \ln\left(\frac{|\Sigma_2|}{|\Sigma_1|}\right)\right), \tag{17}$$

in which $|\cdot|$ is the determinant of matrix.

The CS divergence for $p$ and $q$ is given by (Kampa et al., 2011):

$$D_{\text{CS}}(p; q) = -\log(z_{12}) + \frac{1}{2}\log(z_{11}) + \frac{1}{2}\log(z_{22}), \tag{18}$$

where

$$z_{12} = \mathcal{N}(\mu_1|\mu_2, (\Sigma_1 + \Sigma_2)) = \frac{\exp(-\frac{1}{2}(\mu_1 - \mu_2)^T)(\Sigma_1 + \Sigma_2)^{-1}(\mu_1 - \mu_2)}{\sqrt{(2\pi)^d|\Sigma_1 + \Sigma_2|}}$$

$$z_{11} = \frac{1}{\sqrt{(2\pi)^d|2\Sigma_1|}} \tag{19}$$

$$z_{22} = \frac{1}{\sqrt{(2\pi)^d|2\Sigma_2|}}.$$

Therefore,

$$
\begin{aligned}
D_{\text{CS}}(p; q) &= \frac{1}{2}(\mu_2 - \mu_1)^T(\Sigma_1 + \Sigma_2)^{-1}(\mu_2 - \mu_1) + \log(\sqrt{(2\pi)^d|\Sigma_1 + \Sigma_2|}) - \\
&\quad \frac{1}{2}\log(\sqrt{(2\pi)^d|2\Sigma_1|}) - \frac{1}{2}\log(\sqrt{(2\pi)^d|2\Sigma_2|}) \\
&= \frac{1}{2}(\mu_2 - \mu_1)^T(\Sigma_1 + \Sigma_2)^{-1}(\mu_2 - \mu_1) + \frac{1}{2}\ln\left(\frac{|\Sigma_1 + \Sigma_2|}{2^d\sqrt{|\Sigma_1||\Sigma_2|}}\right).
\end{aligned} \tag{20}
$$

We first consider the difference between $D_{\text{CS}}(p; q)$ and $D_{\text{KL}}(p; q)$ results from mean vector discrepancy, i.e., $\mu_1 - \mu_2$.

**Corollary 1.** *(Horn & Johnson, 2012) For any two positive semi-definite Hermitian matrices $A$ and $B$ of size $n \times n$, then $A - B$ is positive semi-definite if and only if $B^{-1} - A^{-1}$ is also positive semi-definite.*

Applying Corollary 1 to $(\Sigma_1 + \Sigma_2) - \Sigma_2$ (which is positive semi-definite), we obtain that $\Sigma_2^{-1} - (\Sigma_1 + \Sigma_2)^{-1}$ is also positive semi-definite, from which we obtain:

$$
\begin{aligned}
&(\mu_2 - \mu_1)^T\Sigma_2^{-1}(\mu_2 - \mu_1) - (\mu_2 - \mu_1)^T(\Sigma_1 + \Sigma_2)^{-1}(\mu_2 - \mu_1) \\
&= (\mu_2 - \mu_1)^T\left[\Sigma_2^{-1} - (\Sigma_1 + \Sigma_2)^{-1}\right](\mu_2 - \mu_1) \geq 0.
\end{aligned} \tag{21}
$$

We then consider the difference between $D_{\text{CS}}(p; q)$ and $D_{\text{KL}}(p; q)$ results from covariance matrix discrepancy, i.e., $\Sigma_1 - \Sigma_2$.

We have,

$$
\begin{aligned}
2(D_{\text{CS}}(p;q) - D_{\text{KL}}(p;q))_{\mu_1=\mu_2} &= \log\left(\frac{|\Sigma_1 + \Sigma_2|}{2^d\sqrt{|\Sigma_1||\Sigma_2|}}\right) - \log\left(\frac{|\Sigma_2|}{|\Sigma_1|}\right) - \text{tr}(\Sigma_2^{-1}\Sigma_1) + d. \\
&= -d\log 2 + \log\left(|\Sigma_1 + \Sigma_2|\right) - \frac{1}{2}(\log|\Sigma_1| + \log|\Sigma_2|) \\
&\quad - \log|\Sigma_2| + \log|\Sigma_1| - \text{tr}(\Sigma_2^{-1}\Sigma_1) + d \\
&= -d\log 2 + \log\left(\frac{|\Sigma_1 + \Sigma_2|}{|\Sigma_2|}\right) + \frac{1}{2}\log\left(\frac{|\Sigma_1|}{|\Sigma_2|}\right) - \text{tr}(\Sigma_2^{-1}\Sigma_1) + d \\
&= -d\log 2 + \log\left(|\Sigma_2^{-1}\Sigma_1 + I|\right) + \frac{1}{2}\log\left(|\Sigma_2^{-1}\Sigma_1|\right) - \text{tr}(\Sigma_2^{-1}\Sigma_1) + d
\end{aligned}
$$
(22)

Let $\{\lambda_i\}_{i=1}^d$ denote the eigenvalues of $\Sigma_2^{-1}\Sigma_1$, which are non-negative, since $\Sigma_2^{-1}\Sigma_1$ is also positive semi-definite.

We have:

$$
|\Sigma_2^{-1}\Sigma_1| = \left[\left(\prod_{i=1}^d \lambda_i\right)^{1/d}\right]^d \leq \left[\frac{1}{d}\sum_{i=1}^d \lambda_i\right]^d = \left(\frac{1}{d}\text{tr}(\Sigma_2^{-1}\Sigma_1)\right)^d,
$$
(23)

in which we use the property that geometric mean is no greater than the arithmetic mean.

Similarly, we have:

$$
|\Sigma_2^{-1}\Sigma_1 + I| = \prod_{i=1}^d (1 + \lambda_i) \leq \left[\frac{1}{d}\sum_{i=1}^d (1 + \lambda_i)\right]^d = \left(1 + \frac{1}{d}\text{tr}(\Sigma_2^{-1}\Sigma_1)\right)^d.
$$
(24)

By plugging Eqs. (23) and (24) into Eq. (22), we obtain:

$$
\begin{aligned}
2(D_{\text{CS}}(p;q) - D_{\text{KL}}(p;q))_{\mu_1=\mu_2} &= -d\log 2 + \log\left(|\Sigma_2^{-1}\Sigma_1 + I|\right) + \frac{1}{2}\log\left(|\Sigma_2^{-1}\Sigma_1|\right) - \text{tr}(\Sigma_2^{-1}\Sigma_1) + d \\
&\leq -d\log 2 + d\log(1 + \frac{1}{d}\text{tr}(\Sigma_2^{-1}\Sigma_1)) + \frac{d}{2}\log(\frac{1}{d}\text{tr}(\Sigma_2^{-1}\Sigma_1)) - \text{tr}(\Sigma_2^{-1}\Sigma_1) + d.
\end{aligned}
$$
(25)

Let us denote $x = \text{tr}(\Sigma_2^{-1}\Sigma_1) = \sum_{i=1}^d \lambda_i \geq 0$, then

$$
2(D_{\text{CS}}(p;q) - D_{\text{KL}}(p;q))_{\mu_1=\mu_2} = f(x) = -d\log 2 + d\log(1 + \frac{x}{d}) + \frac{d}{2}\log(\frac{x}{d}) - x + d. \quad (26)
$$

Let $f'(x) = 0$, we have $x = d$. Since $f''(x = d) < 0$, we have,

$$
2(D_{\text{CS}}(p;q) - D_{\text{KL}}(p;q))_{\mu_1=\mu_2} = f(x) \leq f(x = d) = 0. \quad (27)
$$

Combining Eq. (21) with Eq. (27), we obtain:

$$
D_{\text{CS}}(p;q) - D_{\text{KL}}(p;q) \leq 0. \quad (28)
$$

The above analysis also applies to $D_{\text{KL}}(q;p)$.

Specifically, we have:

$$
2(D_{\text{KL}}(q;p) - D_{\text{CS}}(p;q))_{\Sigma_1=\Sigma_2} = (\mu_2 - \mu_1)^T\left[\Sigma_1^{-1} - (\Sigma_1 + \Sigma_2)^{-1}\right](\mu_2 - \mu_1) \geq 0, \quad (29)
$$

$$
2(D_{\text{CS}}(p;q) - D_{\text{KL}}(q;p))_{\mu_1=\mu_2} = \log\left(\frac{|\Sigma_1 + \Sigma_2|}{2^d\sqrt{|\Sigma_1||\Sigma_2|}}\right) - \log\left(\frac{|\Sigma_1|}{|\Sigma_2|}\right) - \text{tr}(\Sigma_1^{-1}\Sigma_2) + d.
$$

$$
= -d\log 2 + \log(|\Sigma_1 + \Sigma_2|) - \frac{1}{2}(\log|\Sigma_1| + \log|\Sigma_2|)
$$

$$
- \log|\Sigma_1| + \log|\Sigma_2| - \text{tr}(\Sigma_1^{-1}\Sigma_2) + d
$$

$$
= -d\log 2 + \log\left(\frac{|\Sigma_1 + \Sigma_2|}{|\Sigma_1|}\right) + \frac{1}{2}\log\left(\frac{|\Sigma_2|}{|\Sigma_1|}\right) - \text{tr}(\Sigma_1^{-1}\Sigma_2) + d
$$

$$
= -d\log 2 + \log\left(|\Sigma_1^{-1}\Sigma_2 + I|\right) + \frac{1}{2}\log\left(|\Sigma_1^{-1}\Sigma_2|\right) - \text{tr}(\Sigma_1^{-1}\Sigma_2) + d
$$

$$
\leq -d\log 2 + d\log(1 + \frac{1}{d}\text{tr}(\Sigma_1^{-1}\Sigma_2)) + \frac{d}{2}\log(\frac{1}{d}\text{tr}(\Sigma_1^{-1}\Sigma_2)) - \text{tr}(\Sigma_1^{-1}\Sigma_2) + d
$$

$$
\leq 0.
$$

$$(30)$$

Hence,

$$
D_{\text{CS}}(p;q) - D_{\text{KL}}(q;p) \leq 0. \tag{31}
$$

Combining Eq. (28) and Eq. (31), we obtain:

$$
D_{\text{CS}}(p;q) \leq \min\{D_{\text{KL}}(p;q), D_{\text{KL}}(q;p)\}. \tag{32}
$$

$\square$

## A.2 PROOF OF THE PROPOSITION 2

**Proposition 2** (Empirical Estimator of $D_{\text{CS}}(\mathbb{P}_{\text{f}}(\mathbf{s}_{t+1}|\mathbf{s}_t, \mathbf{a}_t); \mathbb{P}_{\text{c}}(\mathbf{s}_{t+1}|\mathbf{s}_t, \mathbf{a}_t))$ (Yu et al., 2023)).
*Given observations in the $2\tau$-length trajectory fraction $\mathcal{T} = \{[(\mathbf{s}_{t+1})_i, \{\mathbf{s}_t, \mathbf{a}_t\}_i]\}_{i=1}^{2\tau}$, dividing them into two fractions such that $\{[(\mathbf{s}_{t+1})_i, \{\mathbf{s}_t, \mathbf{a}_t\}_i]\}_{i=1}^{\tau}$ are sampled from distribution $\mathbb{P}_{\text{f}}(\mathbf{s}_{t+1}, \{\mathbf{s}_t, \mathbf{a}_t\})$ and $\{[(\mathbf{s}_{t+1})_i, \{\mathbf{s}_t, \mathbf{a}_t\}_i]\}_{i=\tau+1}^{2\tau}$ are sampled from $\mathbb{P}_{\text{c}}(\mathbf{s}_{t+1}, \{\mathbf{s}_t, \mathbf{a}_t\})$). Let $K^{\text{f}}$ and $L^{\text{f}}$ denote, respectively, the Gram matrices[3] for the concatenated variable $\{\mathbf{s}_t, \mathbf{a}_t\}$ and the variable $\mathbf{s}_{t+1}$ in the distribution $\mathbb{P}_{\text{f}}$. That is, $\left(K^{\text{f}}\right)_{ij} = \kappa(\{\mathbf{s}_t, \mathbf{a}_t\}_i - \{\mathbf{s}_t, \mathbf{a}_t\}_j)$, $\left(L^{\text{f}}\right)_{ij} = \kappa(\{\mathbf{s}_{t+1}\}_i - \{\mathbf{s}_{t+1}\}_j)$ for $i, j = 1 : \tau$, in which $\kappa$ is a Gaussian kernel and takes the form of $\kappa = \exp\left(-\frac{\|\cdot\|^2}{2\sigma^2}\right)$. Similarly, let $K^{\text{c}}$ and $L^{\text{c}}$ denote, respectively, the Gram matrices for the variable $\{\mathbf{s}_t, \mathbf{a}_t\}$ and the variable $\mathbf{s}_{t+1}$ in the distribution $\mathbb{P}_{\text{c}}$. Meanwhile, let $K^{\text{fc}} \in \mathbb{R}^{\tau \times \tau}$ (i.e., $\left(K^{\text{fc}}\right)_{ij} = \kappa(\{\mathbf{s}_t, \mathbf{a}_t\}_i - \{\mathbf{s}_t, \mathbf{a}_t\}_j)$, $i = 1 : \tau$ and $j = \tau + 1 : 2\tau$) denote the Gram matrix for variable $\{\mathbf{s}_t, \mathbf{a}_t\}$ from distribution $\mathbb{P}_{\text{f}}$ to distribution $\mathbb{P}_{\text{c}}$, and $L^{\text{fc}} \in \mathbb{R}^{\tau \times \tau}$ the Gram matrix for variable $\mathbf{s}_{t+1}$ from distribution $\mathbb{P}_{\text{f}}$ to distribution $\mathbb{P}_{\text{c}}$. The Gram matrices $K^{\text{cf}}$ and $L^{\text{cf}}$ can be defined similarly. The empirical estimation of $D_{\text{CS}}(\mathbb{P}_{\text{f}}(\mathbf{s}_{t+1}|\mathbf{s}_t, \mathbf{a}_t); \mathbb{P}_{\text{c}}(\mathbf{s}_{t+1}|\mathbf{s}_t, \mathbf{a}_t))$ is given by:*

$$
\widehat{D}_{CS}(\mathbb{P}_{\text{f}}(\mathbf{s}_{t+1}|\mathbf{s}_t, \mathbf{a}_t); \mathbb{P}_{\text{c}}(\mathbf{s}_{t+1}|\mathbf{s}_t, \mathbf{a}_t)) = \log\left(\sum_{j=1}^{\tau}\left(\frac{\sum_{i=1}^{\tau} K_{ji}^{\text{f}}L_{ji}^{\text{f}}}{(\sum_{i=1}^{\tau} K_{ji}^{\text{f}})^2}\right)\right) + \log\left(\sum_{j=1}^{\tau}\left(\frac{\sum_{i=1}^{\tau} K_{ji}^{\text{c}}L_{ji}^{\text{c}}}{(\sum_{i=1}^{\tau} K_{ji}^{\text{c}})^2}\right)\right)
$$

$$
- \log\left(\sum_{j=1}^{\tau}\left(\frac{\sum_{i=1}^{\tau} K_{ji}^{\text{fc}}L_{ji}^{\text{fc}}}{(\sum_{i=1}^{\tau} K_{ji}^{\text{c}})(\sum_{i=1}^{\tau} K_{ji}^{\text{fc}})}\right)\right) - \log\left(\sum_{j=1}^{\tau}\left(\frac{\sum_{i=1}^{\tau} K_{ji}^{\text{cf}}L_{ji}^{\text{cf}}}{(\sum_{i=1}^{\tau} K_{ji}^{\text{cf}})(\sum_{i=1}^{\tau} K_{\text{cf}}^{\text{f}})}\right)\right).
$$

$$(33)$$

---

[3]In kernel learning, the Gram or kernel matrix is a symmetric matrix where each entry is the inner product of the corresponding data points in a reproducing kernel Hilbert space (RKHS), defined by kernel function $\kappa$.

Meanwhile, recall the conditional Cauchy-Schwarz divergence (CCSD) for MaxCondDiv:

$$D_{\mathrm{CS}}(\mathbb{P}_{\mathrm{f}}(\mathbf{s}_{t+1}|\mathbf{s}_t, \mathbf{a}_t); \mathbb{P}_{\mathrm{c}}(\mathbf{s}_{t+1}|\mathbf{s}_t, \mathbf{a}_t))$$

$$= -2\log\left(\int_{\mathcal{S}_{t+1}} \int_{\{\mathcal{S}_t, \mathcal{A}_t\}} \frac{\mathbb{P}_{\mathrm{f}}(\mathbf{s}_{t+1}, \{\mathbf{s}_t, \mathbf{a}_t\}) \mathbb{P}_{\mathrm{c}}(\mathbf{s}_{t+1}, \{\mathbf{s}_t, \mathbf{a}_t\})}{\mathbb{P}_{\mathrm{f}}(\{\mathbf{s}_t, \mathbf{a}_t\}) \mathbb{P}_{\mathrm{c}}(\{\mathbf{s}_t, \mathbf{a}_t\})} d\{\mathbf{s}_t, \mathbf{a}_t\} d\mathbf{s}_{t+1}\right)$$

$$+ \log\left(\int_{\mathcal{S}_{t+1}} \int_{\{\mathcal{S}_t, \mathcal{A}_t\}} \frac{\mathbb{P}_{\mathrm{f}}^2(\mathbf{s}_{t+1}, \{\mathbf{s}_t, \mathbf{a}_t\})}{\mathbb{P}_{\mathrm{f}}^2(\{\mathbf{s}_t, \mathbf{a}_t\})} d\{\mathbf{s}_t, \mathbf{a}_t\} d\mathbf{s}_{t+1}\right) \qquad (34)$$

$$+ \log\left(\int_{\mathcal{S}_{t+1}} \int_{\{\mathcal{S}_t, \mathcal{A}_t\}} \frac{\mathbb{P}_{\mathrm{c}}^2(\mathbf{s}_{t+1}, \{\mathbf{s}_t, \mathbf{a}_t\})}{\mathbb{P}_{\mathrm{c}}^2(\{\mathbf{s}_t, \mathbf{a}_t\})} d\{\mathbf{s}_t, \mathbf{a}_t\} d\mathbf{s}_{t+1}\right),$$

where $\mathcal{S}_{t+1}, \{\mathcal{S}_t, \mathcal{A}_t\}$ are world set of $\{\mathbf{s}_{t+1}, \{\mathbf{s}_t, \mathbf{a}_t\}\}$, respectively.

*Proof.* We first demonstrate how to estimate the two conditional quadratic terms (i.e., $\int_{\mathcal{X}} \int_{\mathcal{Y}} \frac{p^2(\mathbf{x}, \mathbf{y})}{p^2(\mathbf{x})} d\mathbf{x} d\mathbf{y}$):

$$\int_{\mathcal{X}} \int_{\mathcal{Y}} \frac{p^2(\mathbf{x}, \mathbf{y})}{p^2(\mathbf{x})} d\mathbf{x} d\mathbf{y} = \mathbb{E}_{p(X,Y)}\left[\frac{p(X, Y)}{p^2(X)}\right] \approx \frac{1}{\tau} \sum_{j=1}^{\tau} \frac{p(\mathbf{x}_j, \mathbf{y}_j)}{p^2(\mathbf{x}_j)}. \qquad (35)$$

By kernel density estimator (KDE), we have:

$$\frac{p(\mathbf{x}_j, \mathbf{y}_j)}{p^2(\mathbf{x}_j)} \approx \tau \frac{\sum_{i=1}^{\tau} \kappa_\sigma(\mathbf{x}_j^p - \mathbf{x}_i^p) \kappa_\sigma(\mathbf{y}_j^p - \mathbf{y}_i^p)}{\left(\sum_{i=1}^{M} \kappa_\sigma(\mathbf{x}_j^p - \mathbf{x}_i^p)\right)^2}. \qquad (36)$$

where $\kappa$ denotes Gaussian kernel; $\sigma$ denotes kernel size. Therefore:

$$\int_{\mathcal{X}} \int_{\mathcal{Y}} \frac{p^2(\mathbf{x}, \mathbf{y})}{p^2(\mathbf{x})} d\mathbf{x} d\mathbf{y} \approx \sum_{j=1}^{\tau} \left(\frac{\sum_{i=1}^{M} \kappa_\sigma(\mathbf{x}_j^p - \mathbf{x}_i^p) \kappa_\sigma(\mathbf{y}_j^p - \mathbf{y}_i^p)}{\left(\sum_{i=1}^{\tau} \kappa_\sigma(\mathbf{x}_j^p - \mathbf{x}_i^p)\right)^2}\right). \qquad (37)$$

Similarly, :

$$\int_{\mathcal{X}} \int_{\mathcal{Y}} \frac{q^2(\mathbf{x}, \mathbf{y})}{q^2(\mathbf{x})} d\mathbf{x} d\mathbf{y} \approx \sum_{j=1}^{\tau} \left(\frac{\sum_{i=1}^{\tau} \kappa_\sigma(\mathbf{x}_j^q - \mathbf{x}_i^q) \kappa_\sigma(\mathbf{y}_j^q - \mathbf{y}_i^q)}{\left(\sum_{i=1}^{\tau} \kappa_\sigma(\mathbf{x}_j^q - \mathbf{x}_i^q)\right)^2}\right). \qquad (38)$$

Then we demonstrate how to estimate the cross term. i.e., $\int_{\mathcal{X}} \int_{\mathcal{Y}} \frac{p(\mathbf{x}, \mathbf{y}) q(\mathbf{x}, \mathbf{y})}{p(\mathbf{x}) q(\mathbf{x})} d\mathbf{x} d\mathbf{y}$, in a similar way:

$$\int_{\mathcal{X}} \int_{\mathcal{Y}} \frac{p(\mathbf{x}, \mathbf{y}) q(\mathbf{x}, \mathbf{y})}{p(\mathbf{x}) q(\mathbf{x})} d\mathbf{x} d\mathbf{y} = \mathbb{E}_{p(X,Y)}\left[\frac{q(X, Y)}{p(X) q(X)}\right] \approx \frac{1}{\tau} \sum_{j=1}^{\tau} \frac{q(\mathbf{x}_j, \mathbf{y}_j)}{p(\mathbf{x}_j) q(\mathbf{x}_j)}. \qquad (39)$$

And then by KDE, we have:

$$\int_{\mathcal{X}} \int_{\mathcal{Y}} \frac{p(\mathbf{x}, \mathbf{y}) q(\mathbf{x}, \mathbf{y})}{p(\mathbf{x}) q(\mathbf{x})} d\mathbf{x} d\mathbf{y} \approx \sum_{j=1}^{\tau} \left(\frac{\sum_{i=1}^{\tau} \kappa_\sigma(\mathbf{x}_j^p - \mathbf{x}_i^q) \kappa_\sigma(\mathbf{y}_j^p - \mathbf{y}_i^q)}{\sum_{i=1}^{\tau} \kappa_\sigma(\mathbf{x}_j^p - \mathbf{x}_i^p) \sum_{i=1}^{\tau} \kappa_\sigma(\mathbf{x}_j^p - \mathbf{x}_i^q)}\right). \qquad (40)$$

Similarly, we can also empirically estimate $\int_{\mathcal{X}} \int_{\mathcal{Y}} \frac{p(\mathbf{x}, \mathbf{y}) q(\mathbf{x}, \mathbf{y})}{p(\mathbf{x}) q(\mathbf{x})} dx dy$ over $q(\mathbf{x}, \mathbf{y})$:

$$\int_{\mathcal{X}} \int_{\mathcal{Y}} \frac{p(\mathbf{x}, \mathbf{y}) q(\mathbf{x}, \mathbf{y})}{p(\mathbf{x}) q(\mathbf{x})} d\mathbf{x} d\mathbf{y} \approx \sum_{j=1}^{\tau} \left(\frac{\sum_{i=1}^{\tau} \kappa_\sigma(\mathbf{x}_j^q - \mathbf{x}_i^p) \kappa_\sigma(\mathbf{y}_j^q - \mathbf{y}_i^p)}{\sum_{i=1}^{\tau} \kappa_\sigma(\mathbf{x}_j^q - \mathbf{x}_i^p) \sum_{i=1}^{\tau} \kappa_\sigma(\mathbf{x}_j^q - \mathbf{x}_i^q)}\right). \qquad (41)$$

Finally, we represent the equation above in the form of Gram matrices. Let $K^{\mathrm{f}}$ and $L^{\mathrm{f}}$ denote, respectively, the Gram matrices for the input variable $\mathbf{x}$ and output variable $\mathbf{y}$ in the distribution $p$ (i.e., $\mathbb{P}_{\mathrm{f}}$). Further, let $(K)_{ji}$ denotes the $(j, i)$-th element of a matrix $K$ (i.e., the $j$-th row and $i$-th column of $K$). We have:

$$\int_{\mathcal{S}_{t+1}} \int_{\{\mathcal{S}_t, \mathcal{A}_t\}} \frac{\mathbb{P}_{\mathrm{f}}^2(\mathbf{s}_{t+1}, \{\mathbf{s}_t, \mathbf{a}_t\})}{\mathbb{P}_{\mathrm{f}}^2(\{\mathbf{s}_t, \mathbf{a}_t\})} d\{\mathbf{s}_t, \mathbf{a}_t\} d\mathbf{s}_{t+1} = \int_{\mathcal{X}} \int_{\mathcal{Y}} \frac{p^2(\mathbf{x}, \mathbf{y})}{p^2(\mathbf{x})} d\mathbf{x} d\mathbf{y} \approx \sum_{j=1}^{\tau} \left(\frac{\sum_{i=1}^{\tau} K_{ji}^{\mathrm{f}} L_{ji}^{\mathrm{f}}}{(\sum_{i=1}^{\tau} K_{ji}^{\mathrm{f}})^2}\right).$$

$$(42)$$

Similarly, let $K^c$ and $L^c$ denote, respectively, the Gram matrices for input variable $\mathbf{x}$ and output variable $\mathbf{y}$ in the distribution $q$ (i.e., $\mathbb{P}_c$). We have:

$$\int_{\mathcal{S}_{t+1}} \int_{\{\mathcal{S}_t, \mathcal{A}_t\}} \frac{\mathbb{P}_c^2(\mathbf{s}_{t+1}, \{\mathbf{s}_t, \mathbf{a}_t\})}{\mathbb{P}_c^2(\{\mathbf{s}_t, \mathbf{a}_t\})} d\{\mathbf{s}_t, \mathbf{a}_t\} d\mathbf{s}_{t+1} = \int_{\mathcal{X}} \int_{\mathcal{Y}} \frac{q^2(\mathbf{x}, \mathbf{y})}{q^2(\mathbf{x})} d\mathbf{x} d\mathbf{y} \approx \sum_{j=1}^{\tau} \left( \frac{\sum_{i=1}^{\tau} K_{ji}^c L_{ji}^c}{(\sum_{i=1}^{\tau} K_{ji}^c)^2} \right).$$
(43)

Further, let $K^{fc} \in \mathbb{R}^{\tau \times \tau}$ denote the Gram matrix between distributions $p$ and $q$ for input variable $\mathbf{x}$, and $L^{fc}$ the Gram matrix between distributions $p$ and $q$ for output variable $\mathbf{y}$. According to Eq. (40), we have:

$$\int_{\mathcal{S}_{t+1}} \int_{\{\mathcal{S}_t, \mathcal{A}_t\}} \frac{\mathbb{P}_f(\mathbf{s}_{t+1}, \{\mathbf{s}_t, \mathbf{a}_t\})\mathbb{P}_c(\mathbf{s}_{t+1}, \{\mathbf{s}_t, \mathbf{a}_t\})}{\mathbb{P}_f(\{\mathbf{s}_t, \mathbf{a}_t\})\mathbb{P}_c(\{\mathbf{s}_t, \mathbf{a}_t\})} d\{\mathbf{s}_t, \mathbf{a}_t\} d\mathbf{s}_{t+1} = \int_{\mathcal{X}} \int_{\mathcal{Y}} \frac{p(\mathbf{x}, \mathbf{y})q(\mathbf{x}, \mathbf{y})}{p(\mathbf{x})q(\mathbf{x})} d\mathbf{x} d\mathbf{y}$$
$$\approx \sum_{j=1}^{\tau} \left( \frac{\sum_{i=1}^{\tau} K_{ji}^{fc} L_{ji}^{fc}}{(\sum_{i=1}^{\tau} K_{ji}^c)(\sum_{i=1}^{\tau} K_{ji}^{fc})} \right).$$
(44)

Similarly, according to Eq. 41, we have

$$\log \left( \int_{\mathcal{S}_{t+1}} \int_{\{\mathcal{S}_t, \mathcal{A}_t\}} \frac{\mathbb{P}_f(\mathbf{s}_{t+1}, \{\mathbf{s}_t, \mathbf{a}_t\})\mathbb{P}_c(\mathbf{s}_{t+1}, \{\mathbf{s}_t, \mathbf{a}_t\})}{\mathbb{P}_f(\{\mathbf{s}_t, \mathbf{a}_t\})\mathbb{P}_c(\{\mathbf{s}_t, \mathbf{a}_t\})} d\{\mathbf{s}_t, \mathbf{a}_t\} d\mathbf{s}_{t+1} \right)$$
$$\approx \log \left( \sum_{j=1}^{\tau} \left( \frac{\sum_{i=1}^{\tau} K_{ji}^{cf} L_{ji}^{cf}}{(\sum_{i=1}^{\tau} K_{ji}^{cf})(\sum_{i=1}^{\tau} K_{cf}^f)} \right) \right).$$
(45)

Therefore, we have:

$$\widehat{D}_{CS}(\mathbb{P}_f(\mathbf{s}_{t+1}|\mathbf{s}_t, \mathbf{a}_t); \mathbb{P}_c(\mathbf{s}_{t+1}|\mathbf{s}_t, \mathbf{a}_t)) = \log \left( \sum_{j=1}^{\tau} \left( \frac{\sum_{i=1}^{\tau} K_{ji}^f L_{ji}^f}{(\sum_{i=1}^{\tau} K_{ji}^f)^2} \right) \right) + \log \left( \sum_{j=1}^{\tau} \left( \frac{\sum_{i=1}^{\tau} K_{ji}^c L_{ji}^c}{(\sum_{i=1}^{\tau} K_{ji}^c)^2} \right) \right)$$
$$- \log \left( \sum_{j=1}^{\tau} \left( \frac{\sum_{i=1}^{\tau} K_{ji}^{fc} L_{ji}^{fc}}{(\sum_{i=1}^{\tau} K_{ji}^c)(\sum_{i=1}^{\tau} K_{ji}^{fc})} \right) \right) - \log \left( \sum_{j=1}^{\tau} \left( \frac{\sum_{i=1}^{\tau} K_{ji}^{cf} L_{ji}^{cf}}{(\sum_{i=1}^{\tau} K_{ji}^{cf})(\sum_{i=1}^{\tau} K_{cf}^f)} \right) \right)$$
(46)

$\square$

### A.3 PROOF OF THE PROPOSITION 3

**Proposition 3.** *Let $X$ and $Y$ be two random variable with marginal PDFs $p_X(x) = p_X(X = x)$ and $p_Y(y) = p_Y(Y = y)$, respectively, where $x \in \mathcal{R}$ and $y \in \mathcal{R}$. Let $p_{XY}(x, y) = p_{XY}(X = x, Y = y)$ denotes the joint PDF. We have:*

$$\frac{1}{2} H_2(x) + \frac{1}{2} H_2(y) - I_2(x, y) \geq D_{cs}(p_X; p_Y)$$
(47)

*iff:*

$$\int_{r \in \mathcal{R}} p_X(X = r) p_Y(Y = r) dr \geq \int_{y \in \mathcal{R}} \int_{x \in \mathcal{R}} p_{XY}^2(x, y) dx dy$$
(48)

*where $H_2(\cdot)$ and $I_2(\cdot, \cdot)$ are 2nd-order Rényi entropy and mutual information.*

To facilitate comprehension of Eq. (48), we provide a visualization in Fig. 7 using a straightforward discrete example involving three possible states within $\mathcal{R}$. The right-hand side of Equation (48) corresponds to the summation of the squared elements in the table. The left-hand side of Eq. (48) is equivalent to the sum of the products of the marginal probabilities. i.e., $p_X(X = r) = \int_{y \in \mathcal{R}} p_{XY}(X = r, Y = y) dy$ and $p_Y(Y = r) = \int_{x \in \mathcal{R}} p_{XY}(X = x, Y = r) dx$.

*Proof.* We first rewrite the left side of Eq. (47):

$$
\begin{aligned}
\frac{1}{2}H_2(x) + \frac{1}{2}H_2(y) - I_2(x,y) &= \frac{1}{2}H_2(x) + \frac{1}{2}H_2(y) - [H_2(x,y) + H_2(x) + H_2(y)] \\
&= H_2(x,y) - \frac{1}{2}H_2(p_X) - \frac{1}{2}H_2(p_Y) \\
&= -\log \int_{y\in\mathcal{R}}\int_{x\in\mathcal{R}} p_{XY}^2(x,y)dxdy - \frac{1}{2}H_2(p_X) - \frac{1}{2}H_2(p_Y)
\end{aligned}
$$

(49)

where $H_2(x,y)$ is the joint entropy. Then we rewrite the right side of Eq. (47):

$$
\begin{aligned}
D_{\mathrm{CS}}(p_X; p_Y) &= -\frac{1}{2}\log \frac{(\int_{r\in\mathcal{R}} p_X(X=r)p_Y(Y=r)dr)^2}{(\int_{r\in\mathcal{R}} p_X^2(X=r)dr)(\int_{r\in\mathcal{R}} p_Y^2(Y=r)dr)} \\
&= -\log(\int_{r\in\mathcal{R}} p_X(X=r)p_Y(Y=r)dr) - \frac{1}{2}H_2(p_X) - \frac{1}{2}H_2(p_Y)
\end{aligned}
$$

(50)

where $-\log(\int_{r\in\mathcal{R}} p_X(X=r)p_Y(Y=r)dr)$ is also the 2-order Rényi cross entropy between $p$ and $q$. Therefore,

$$
\begin{aligned}
&\frac{1}{2}H_2(x) + \frac{1}{2}H_2(y) - I_2(x,y) \geq D_{cs}(p_X; p_Y) \\
\Leftrightarrow &-\log \int_{y\in\mathcal{R}}\int_{x\in\mathcal{R}} p_{XY}^2(x,y)dxdy \geq -\log(\int_{r\in\mathcal{R}} p_X(X=r)p_Y(Y=r)dr) \\
\Leftrightarrow &\int_{r\in\mathcal{R}} p_X(X=r)p_Y(Y=r)dr \geq \int_{y\in\mathcal{R}}\int_{x\in\mathcal{R}} p_{XY}^2(x,y)dxdy
\end{aligned}
$$

(51)

$\square$

Using discrete cases to further elucidate Eq. (48), if we break down both sides of the equation, we find that the left-hand side generates $N^3$ terms, whereas the right-hand side only yields $N^2$ terms, where $N$ represents the number of possible state values in $\mathcal{R}$.

For an arbitrary term on the right-hand side, let's say $p_{XY}^2(4,5)$, there must exist a corresponding term on the left-hand side, specifically $p_{XY}(4,5)\sum_{r=1}^{N} p_{XY}(5, Y=r)$. It's reasonable to assume that $\sum_{r=1}^{N} p_{XY}(5, Y=r) \geq p_{XY}(4,5)$, especially when $N$ is large.

| State | 1 | 2 | 3 |
|-------|-----|-----|-----|
| 1 | p(1,1) | p(1,2) | p(1,3) |
| 2 | p(2,1) | p(2,2) | p(2,3) |
| 3 | p(3,1) | p(3,2) | p(3,3) |

**Figure 7:** Visualization of joint probability table when number of possible states $N = 3$. The circles indicate how to calculate marginal probabilities $p_X$ and $p_Y$, i.e., $p_X(X=r) = \sum_{y=1}^{3} p_{XY}(X=r, Y=y)$ and $p_Y(Y=r) = \sum_{x=1}^{3} p_{XY}(X=x, Y=r)$.

To demonstrate that the of Eq. (48) is a trivial case, we perform a Monte Carlo simulation using a $10 \times 10$ joint probability table. Initially, we assign 100 random numbers from a uniform distribution

in the interval [0, 10] to populate the table. Subsequently, we apply the softmax function to ensure that the sum of the values in the table equals one. This experiment is repeated one million times. In our experiments, the condition holds with a frequency of 99.78%. In Fig. 8, we illiterate some

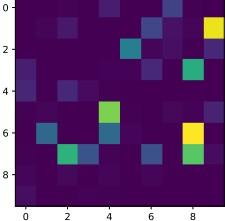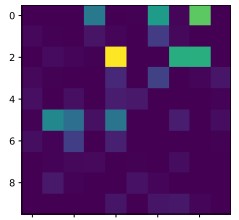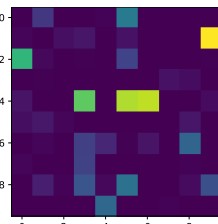

**Figure 8:** Counterexamples of Eq. (48)

counterexamples of Eq. (15). We observe that these joint probability tables often exhibit sparsity, with the probability concentrated on a few non-diagonal elements. In the context of RL environments, this suggests that the agent becomes stuck in a limited number of transitions, which is a rare occurrence, particularly in continuous spaces.

## B    DETAILS OF EXPERIMENTAL SETTING

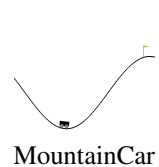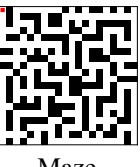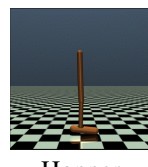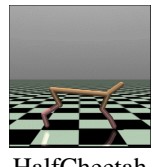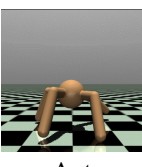

MountainCar        Maze        Hopper        HalfCheetah        Ant

**Figure 9:** Visual interfaces of environments

### B.1    MORE DETAILS OF ENVIRONMENTS

**MountainCar** (Moore, 1990) is a classical deterministic control environment where a car is placed randomly at the bottom of a sinusoidal valley. The available actions for the car are limited to accelerations in either direction.

**Maze** is a classical 2-D space where an agent (red point) is placed at the starting point (0,0). The agent has four available actions, allowing it to move in one of four directions.

**Hopper** (Erez et al., 2012) is a two-dimensional, one-legged agent composed of four main body parts: the torso at the top, the thigh in the middle, the leg at the bottom, and a single foot serving as the base.

**HalfCheetah** (Wawrzyński, 2009) is a 2-dimensional robot comprised of 9 links and 8 joints, including two paws. The objective is to apply torque to the joints and make the cheetah run forward (to the right) as swiftly as possible.

**Ant** (Schulman et al., 2015) is a three-dimensional robot composed of a single torso, which is a freely rotating body, and four legs connected to it. Each leg consists of two links.

In our experiments, observation noise is introduced by rounding state and action values to two decimal places. Fig 9 showcases the visual interfaces of these environments. Table. 1 summarizes the crucial statistics of five environments. We categorize MountainCar and Maze as discrete environments, while Hopper, HalfCheetah, and Ant are classified as continuous environments.

Please note that the default state dimensionality of Ant is 27, but we include the x-y coordinates of the agent by setting the parameter "exclude_current_positions_from_observation" to False. This is consistent with the approach taken in the maximum entropy method described in (Hazan et al., 2019).

| Environment | State Dimensionality | State Space | Action Dimensionality | Action Space |
|---|---|---|---|---|
| MountainCar | 2 | continuous: [-1.2, 0.6] [-0.07, 0.07] | 1 | discrete: {-1,0,1} |
| Maze | 2 | discrete: [20, 20] | 1 | discrete: {0,1,2,3} |
| Hopper | 11 | continuous: [-inf, inf] | 3 | continuous: [-1, 1] |
| HalfCheetah | 17 | continuous: [-inf, inf] | 6 | continuous: [-1, 1] |
| Ant | 29 | continuous: [-inf, inf] | 8 | continuous: [-1, 1] |

**Table 1:** Statistics of Environments

## B.2 INTRODUCTION OF BASELINES

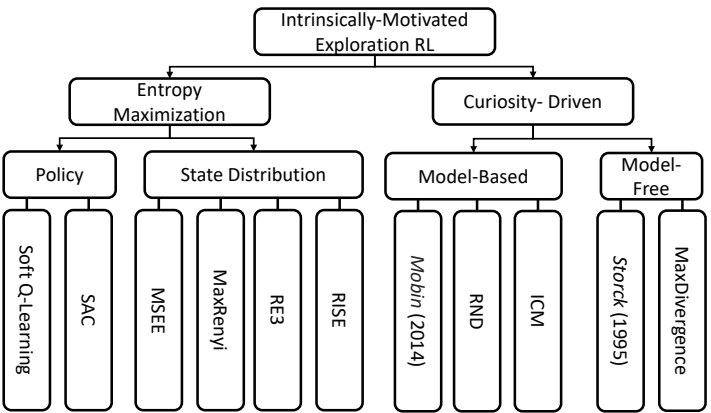

**Figure 10:** The taxonomy diagram depicts different intrinsically motivated exploration RL methods. Our approach, in contrast to most of other curiosity-driven methods, is entirely model-free, meaning it does not require an internal model. Instead, we estimate the divergence between transitions directly from either samples or latent vectors of samples.

Here is a brief introduction to our baselines.In Fig. 10, we illustrate the taxonomy of related intrinsically-motivated exploration RLs.

The **Intrinsic Curiosity Module (ICM)** (Pathak et al., 2017) is an exploration method driven by curiosity. It defines curiosity as the discrepancy between an agent's predicted consequences of its own actions and the actual outcomes. This prediction is made in a visual feature space learned by a self-supervised inverse dynamics model.

The **Random Network Distillation (RND)** (Burda et al., 2019) is an exploration method driven by curiosity. Rather than utilizing prediction errors to measure novelty, this approach predicts the output of a fixed randomly initialized neural network based on the current observation.

The **Provably Efficient Maximum Entropy Exploration (MSEE)** (Hazan et al., 2019) is the pioneering approach that rigorously applies the maximum entropy principle. It provides a proof of policy improvement using the APPROXPLAN/DENSITYEST oracle and directly estimates entropy in observation spaces.

The **Maximum Rényi Entropy Exploration (MaxRényi)** (Zhang et al., 2021) is a Rényi entropy-based variation of MSEE. Besides, it maximizes the entropy in the joint space of action and state.

The **Rényi state entropy maximization (RISE)** (Yuan et al., 2022) is a combination of MSEE and MaxRényi. It replaces Shannon entropy with Rényi entropy and incorporates autoencoders and K-NN methods to estimate entropy in the latent space. RISE emphasizes learning acceleration by leveraging both extrinsic and intrinsic rewards. However, in our experiment, we solely utilize intrinsic rewards.

For MSEE, we perform our experiments using the official codes [4] with the default settings. For MaxRényi, we have adapted the MSEE code by substituting Shannon entropy with Rényi entropy and incorporating the action space when estimating entropies. We have taken this approach because the original code and replication details for MaxRényi are unavailable. For other methods we perform our experiments using the toolbox [5] released by the author of (Yuan et al., 2022).

## B.3 DETAILS OF HYPER-PARAMETER SETTING

For CCSD estimation, there are two hyperparameters to consider: the kernel size $\sigma$ and the buffer size $2\tau$. Table. 2 summarizes our selections of five environments.

| Environments | Thought Experiment | MountainCar | Maze | Hopper | HalfCheetah | Ant |
|---|---|---|---|---|---|---|
| $(\sigma, 2\tau)$ | (1, 50) | (0.1, 400) | (0.1, 50) | (1, 400) | (1, 100) | (2, 50) |

**Table 2:** Hyper-parameters of CCSD estimation

For our Q-learning oracle, we select he discount $\gamma$ to be 0.9. For our Proximal Policy Optimization (PPO) oracle, we summarizes our hyper-parameters in Table. 3

| Hyper-parameters | Value |
|---|---|
| entropy_coef | 1e-2 |
| critic_coef | 0.5 |
| actor_lr | 3e-4 |
| critic_lr | 3e-4 |
| hidden_dim | 256 |
| $\gamma$ | 0.99 |
| $\lambda$ | 0.95 |
| max_clip | 0.2 |
| traj_length | 1000 |
| batch_size | 64 |
| max_grad_norm | 0.5 |
| layer_num | 3 |
| activation_function | torch.tanh |
| last_activation | None |

**Table 3:** Hyper-parameters of PPO

## C ADDITIONAL EXPERIMENTAL RESULTS

### C.1 RESULTS OF ALTERNATIVE ESTIMATOR: CONDITIONAL MAXIMUM MEAN DISCREPANCY (CMMD)

As discussed in the section 3.2 of the paper, a popular choice of conditional divergence estimator is conditional maximum mean discrepancy (CMMD). Here we adopt the estimtor prposed by by Ren *et al.* (Ren et al., 2016). More formally using same notions of proposition 1:

$$\widehat{\text{CMMD}} = \frac{1}{\tau^2} trace(L^{\text{f}} \cdot C^{\text{f}}) + \frac{1}{\tau^2} trace(L^{\text{c}} \cdot C^{\text{c}}) - \frac{2}{\tau^2} trace(L^{\text{fc}} \cdot C^{\text{fc}}), \qquad (52)$$

where $C^{\text{f}} = (K^{\text{f}} + \lambda I)^{-1} K^{\text{f}} (K^{\text{f}} + \lambda I)^{-1}$, $C^{\text{c}} = (K^{\text{c}} + \lambda I)^{-1} K^{\text{c}} (K^{\text{c}} + \lambda I)^{-1}$, $C^{\text{fc}} = (K^{\text{fc}} + \lambda I)^{-1} K^{\text{fc}} (K^{\text{fc}} + \lambda I)^{-1}$, $\lambda$ is a hyper-parameter.

As the equations shown, this method involves taking the matrix inverse of $K + \lambda I$, which is not guaranteed to be invertible. Using alternative computations, such as pseudo-inverses, may lead to imprecise estimations and consequently reduce the stability of learning.

---

[4]https://github.com/abbyvansoest/maxent
[5]https://github.com/yuanmingqi/rl-exploration-baselines

We perform experiments on MountainCar by replacing CCSD with CMMD to evaluate its performance.

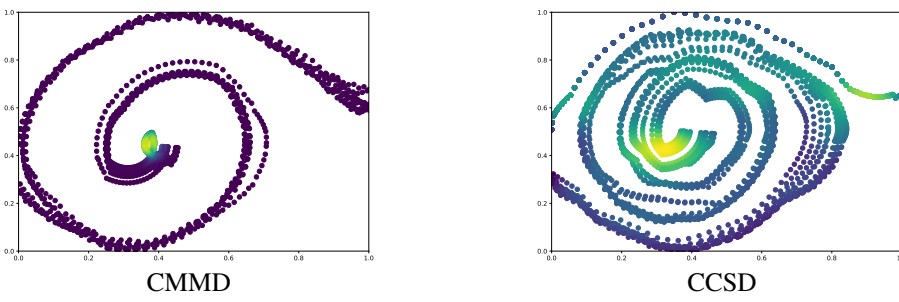

CMMD                                          CCSD

**Figure 11:** Trajectories of different estimators on Mountain Car

Figure 11 displays the traces over 5,000 steps using trained policies obtained from CMMD and CSSD. Despite the possibility of CMMD reaching the flag, CSSD demonstrates better performance overall. The learning curve of CMMD is depicted in Figure 12 (top-left), illustrating its expected instability and difficulty in convergence.

Moreover, CCSD exhibits lower computational complexity compared to CMMD, with $\mathcal{O}(N^2 d)$ versus $\mathcal{O}(N^2 d + N^3)$, where $N$ is the number of samples and $d$ is the dimensionality. In our experiments, CCSD completes 100-episode training in 636.73 seconds, whereas CMMD takes 3326.95 seconds.

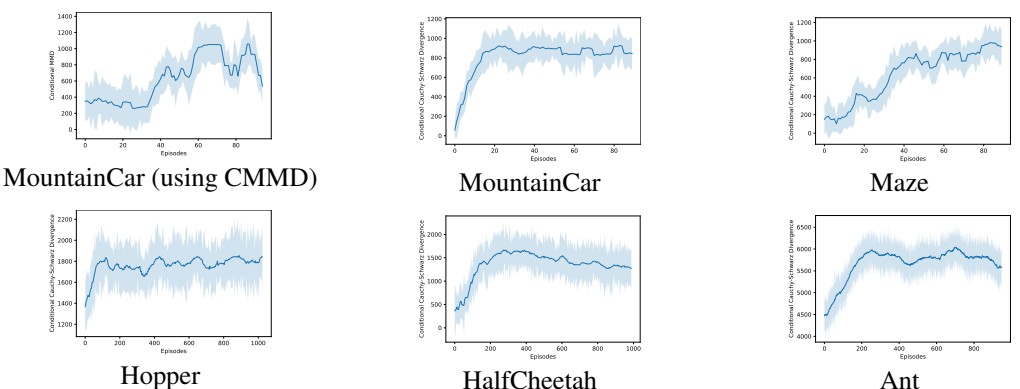

MountainCar (using CMMD)            MountainCar                          Maze

Hopper                                  HalfCheetah                              Ant

**Figure 12:** Learning Curves on five environments. The Y-axis is the cumulative CCSD in current episode.

## C.2 LEARNING CURVES USING KL DIVERGENCE AND CONDITIONAL CAUCHY-SCHWARZ DIVERGENCE

In Fig. 12, we display the learning curves utilizing CCSD as the intrinsic rewards. As training progresses, the CCSD values increase and converge, confirming that the RL agents have learned to achieve higher CCSD values, as intended. It empirically validates the policy improvement of our method.

Conversely, we present learning curves employing conditional KL divergence as intrinsic rewards in Fig. 13. Only the learning curves for HalfCheetah exhibit increases and convergence throughout the learning process. The presence of the term $\log(\frac{p}{q})$ results in NaN (Not-a-Number) values when either $p$ or $q$ becomes very small, leading to instability in the learning process.

## C.3 VIDEOS AND FRAMES RESULTS

Here we illustrated more frames of trained agents on Mujoco in Fig. 14. The agents learn a diverse set of primitive behaviors for all tasks. For half cheetah, they learn skills for flip forwards and backwards at various speeds; ant learns skills for jumping, flipping and Thomas flare maneuvers in many types

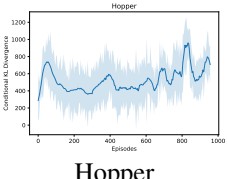
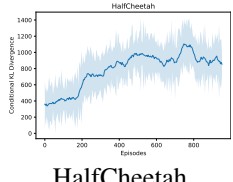
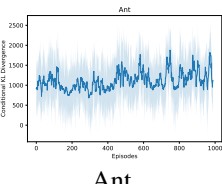

| Hopper | HalfCheetah | Ant |

**Figure 13:** Learning Curves on Mujoco environments. The Y-axis is the cumulative conditional KL divergence in current episode.

of curved trajectories; hopper learns skills for hopping forward and backwards. More videos are attached in the zip file.

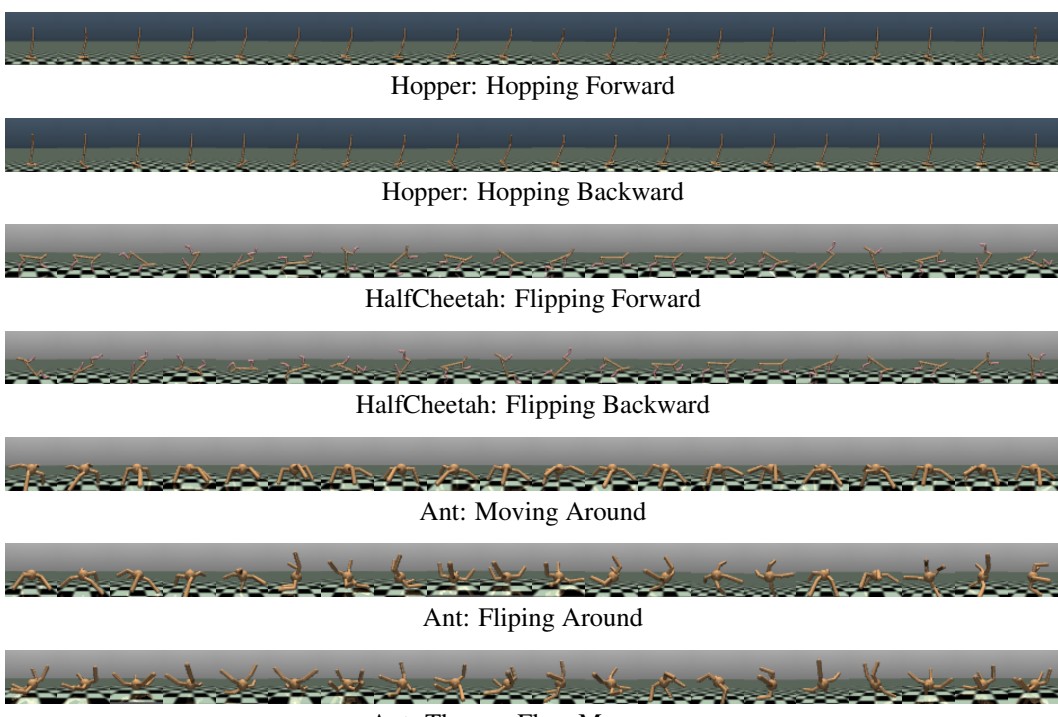

Hopper: Hopping Forward

Hopper: Hopping Backward

HalfCheetah: Flipping Forward

HalfCheetah: Flipping Backward

Ant: Moving Around

Ant: Fliping Around

Ant: Thomas Flare Maneuvers

**Figure 14:** Learned skills without using any extrinsic rewards on Mujoco environments

### C.4 HYPER-PARAMETER ANALYSIS

Within this section, we present results on how hyperparameters selection impacted experimental outcomes. In our MaxCondDiv framework, two pivotal hyperparameters come to the fore: the trajectory fraction length, denoted as $\tau$, and the Gaussian kernel's size, represented as $\sigma$. Due to the constraints of the rebuttal timeframe, our experimentation centers around the two most complex environments featured in our paper: HalfCheetah and Ant.

For the HalfCheetah environment, we explore six distinct values for $2\tau$ and seven values for $\sigma$, specifically [10, 50, 100, 150, 200, 500] and [0.1, 0.5, 0.75, 1, 1.5, 2, 10] respectively. The experiments are conducted 3 times, and the average number of visited states within 10,000 steps using trained policies is reported. As depicted in Fig. 15, MaxCondDiv achieves favorable outcomes when $\tau$ is set between 50 and 200, as well as when $\sigma$ ranges from 0.5 to 1.5. The peak can be readily identified through line search.

For Ant environment, we explore six distinct values for $2\tau$ and seven values for $\sigma$, specifically [10,20,50,100,200,500] and [0.1,0.5,1,2,3,5,10] respectively. Results are similar to HalfCheetah.

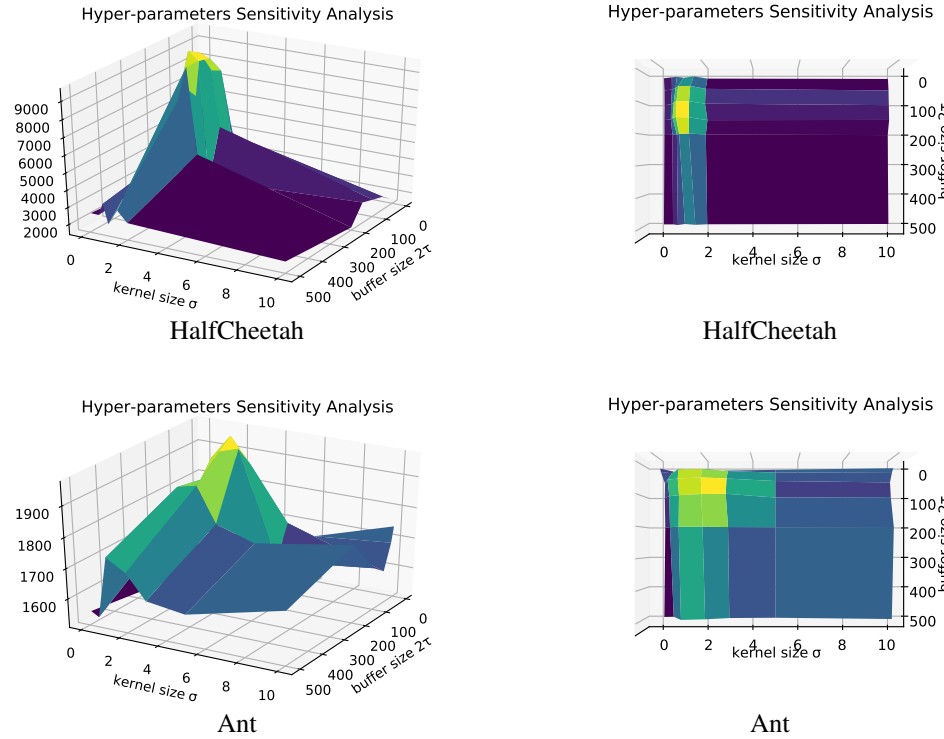

**Figure 15:** How hyper-parameters impact our experimental outcomes on HalfCheetah and Ant. We choose two crucial hyper-parameters, i.e., the trajectory fraction length $\tau$ and the Gaussian kernel size $\sigma$, to construct a 3D surface plot, with the z-axis depicting the number of visited states.

## C.5 DOWNSTREAM TASK: OFFLINE REINFORCEMENT LEARNING RESULTS

Exploration RL has various applications, including the collection of datasets for offline RL (Fu et al., 2020). This process involves two phases. In the exploration phase, we interact with the environment using reward-free exploration RL to collect samples $\{s_{t+1}, a_t, s_t, r(s_t, a_t)\}$ over a significant number of steps. Here, $r(s_t, a_t)$ represents the task-driven extrinsic reward. In the planning phase, we train an offline RL algorithm using the collected dataset, eliminating the need for further interaction with the environments. The results of offline RL can partially indicate the quality of the collected data and, consequently, the effectiveness of the exploration method. We collect datasets for a total of 50 thousands and 20 million steps for discrete environments and Mujoco, respectively, using multiple exploration methods, and utilize the conservative Q-learning (CQL) (Kumar et al., 2020) algorithm as the offline RL method in our experiments.

| Environments | Random | ICM | RND | RISE | MSEE | MaxCondDiv (ours) | Online-Learning (SAC) |
|---|---|---|---|---|---|---|---|
| MountainCar | 0 | 0.454 | 0.548 | 0.575 | 0.521 | **0.587** | 1 |
| Maze | 0 | 0.634 | 0.701 | 0.712 | **0.765** | 0.681 | 1 |
| Hopper | 0.091 | 0.178 | 0.330 | 0.117 | 0.622 | **0.812** | 1 |
| HalfCheetah | **0.375** | 0.185 | 0.322 | 0.239 | 0.259 | 0.262 | 1 |
| Ant | 0.032 | 0.094 | 0.161 | 0.133 | **0.237** | 0.194 | 1 |

**Table 4:** Comparing the results of Random, ICM, RND, RISE, MSEE, and our MaxCondDiv exploration methods, normalized with respect to the online Soft Actor-Critic (SAC) method. The best performance is in bold. The second-best performance is underlined.

Table. 4 summarizes the results on five environments. In the case of MountainCar, all exploration methods perform well and are able to reach the flag within 400 steps. However, our method shows

slightly better performance as it achieves the flag more frequently compared to the other methods. In the case of Maze, the goal is set as starting at coordinates (0, 0) and reaching coordinates (19, 19). All exploration methods show good performance, but MaxEnt outperforms the others by evenly distributing its exploration throughout the entire space, allowing it to include a trajectory to (19, 19) without any bias. In the case of Hopper, our method significantly outperforms the others as it learns to hop forward even without any extrinsic rewards. On the other hand, the other methods mostly result in the agent falling down in the majority of samples. In the case of Hopper, all methods perform worse than the random policy. This is because the agent is intentionally designed to be difficult to flip over when following a random policy (Wawrzyński, 2009). Consequently, the random policy tends to collect samples of sliding forward slowly. However, exploration methods tend to collect samples of flipping over because these states are considered "different" and novel. As a result, in the task of moving forward, these "flipping" samples are noises rather than a useful dataset. In the case of Ant, all methods fail to achieve satisfactory performance. This is because Ant is a complex 3D environment with a large number of possible trajectories. The samples collected by exploration methods do not include enough instances of the desired behavior of "moving forward." As a result, the learned policies based on these exploration methods struggle to navigate and perform well in the Ant environment.

It is important to note that the results obtained are highly influenced by the choice of offline RL algorithm. For instance, if we use batch-constrained Q-learning (Fujimoto et al., 2019), the score of the random policy on HalfCheetah can be as low as $0.014$. On the other hand, if we use offline SAC (Haarnoja et al., 2018), the scores of the random policy may align with the table, but the datasets collected by exploration methods may result in low scores. Therefore, the performance outcomes are strongly tied to the specific offline RL method employed.

Hence, for the sake of simplicity and focus, we choose to use the widely adopted CQL method for our preliminary experiments, as the specific choice of offline RL algorithm is beyond the scope of this paper.

# D   ADDITIONAL DISCUSSION

## D.1   CS DIVERGENCE IS MUCH MORE STABLE THAN KL DIVERGENCE

Apart from Proposition 1, one can also understand the relative stable of CS divergence over conventional KL divergence from another perspective.

For simplicity, let us consider two discrete distributions $p$ and $q$ on the finite set $\mathcal{X} = \{x_1, x_2, \ldots, x_K\}$ (i.e., there are $K$ different discrete states), let us denote $p(x_i) = p(x = x_i)$, we have:

$$
D_{\mathrm{KL}}(p; q) = \sum_{i=1}^{K} p(x_i) \log \left( \frac{p(x_i)}{q(x_i)} \right),
$$
$$
\text{s.t.} \sum_{i=1}^{K} p(x_i) = \sum_{i=1}^{K} q(x_i) = 1.
\tag{53}
$$

$$
D_{\mathrm{CS}}(p; q) = - \log \left( \frac{\sum p(x_i) q(x_i)}{\sqrt{\sum p(x_i)^2} \sqrt{\sum q(x_i)^2}} \right).
\tag{54}
$$

Obviously, when there exists $i$ such that $q(x_i) \to 0$, the resulting $\log \left( \frac{p(x_i)}{q(x_i)} \right) \to \infty$.

CS divergence does not have this issue, this is because (the arithmetic mean is less or equal than the quadratic mean):
$$
\frac{\sum q(x_i)^2}{K} \geq \left( \frac{\sum q(x_i)}{K} \right)^2 = \frac{1}{K^2}.
\tag{55}
$$

Hence,
$$
\sqrt{\sum q(x_i)^2} \geq \sqrt{1/K}.
\tag{56}
$$

That is, the denominator of CS divergence is hardly to reduce to 0, unless $K \approx \infty$ and $q$ (or $p$) is nearly uniform distributed.

### D.2 CLARIFICATION OF PROPOSITION 2

First, we clarify the definition of Gram matrix. Let $X = \{\mathbf{x}_1, \mathbf{x}_2, \ldots, \mathbf{x}_N\}$ be a dataset of $N$ points, each of which is a $d$-dimensional vector, i.e., $\mathbf{x}_i \in \mathcal{R}^d$. The Gram matrix, which is also called the Kernel matrix, is a $N \times N$ symmetric and positive semi-definite matrix where each entry is the inner product of the corresponding data points in the corresponding kernel space, i.e.,

$$G_{ij} = \kappa(\mathbf{x}_i, \mathbf{x}_j) = \langle \phi(\mathbf{x}_i), \phi(\mathbf{x}_j) \rangle. \tag{57}$$

For the linear kernel, the Gram matrix is simply the inner product in input space $G_{ij} = \mathbf{x}_i^T \mathbf{x}_j$. In our paper, we use the most popular Gaussian kernel, so the inner product is measured in a feature space with feature map $\phi$.

Second, we clarify our Proposition 2. Our proposition 2 provides a sample estimator to measure the conditional divergence between $P_f(\mathbf{s}_{t+1}|\mathbf{s}_t, \mathbf{a}_t)$ and $P_c(\mathbf{s}_{t+1}|\mathbf{s}_t, \mathbf{a}_t)$ from two groups of observations: $\{(\mathbf{s}_{t+1})_i, (\mathbf{s}_t, \mathbf{a}_t)_i\}_{i=1}^{\tau}$ (sampled from $P_f$) and $\{(\mathbf{s}_{t+1})_i, (\mathbf{s}_t, \mathbf{a}_t)_i\}_{i=\tau+1}^{2\tau}$ (from $P_c$). Since we are dealing with four variables from two distributions $(\mathbf{s}_{t+1})$; and $(\mathbf{s}_t, \mathbf{a}_t)$ in both $P_f$ and $P_c$. Our proposition implies that, in order to measure the conditional divergence, we need to evaluate six Gram matrix, as shown in Fig. 16:

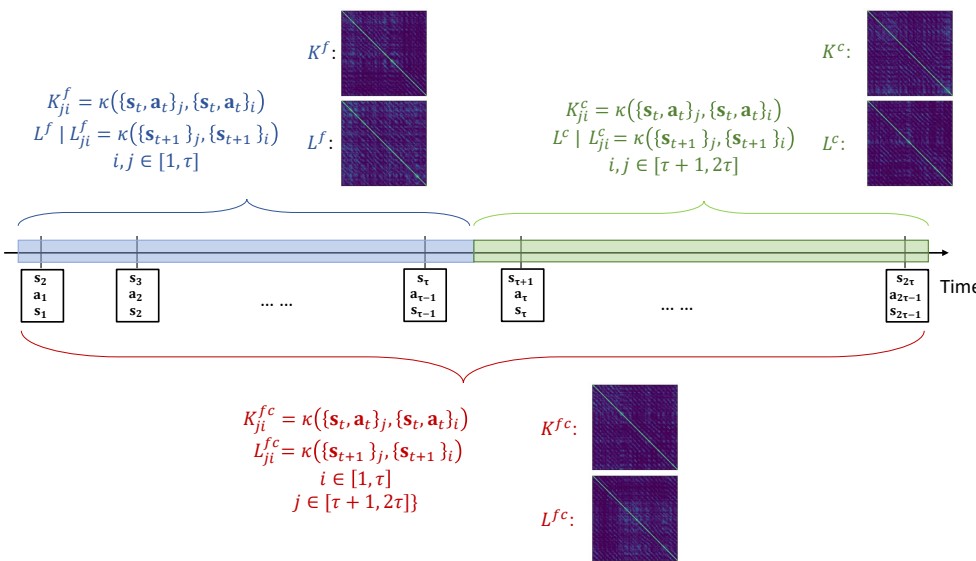

**Figure 16:** Visualization of our computation process of $K^f$, $L^f$, $K^c$, $L^c$, $K^{cf}$ (or $K^{fc}$), $L^{cf}$ (or $L^{fc}$)

1) the Gram matrix $K^f$ which corresponds to concatenated variable $(\mathbf{s}_t, \mathbf{a}_t)$ in distribution $P_f$, i.e., $K^f_{ji} = \kappa(\{\mathbf{s}_t, \mathbf{a}_t\}_j, \{\mathbf{s}_t, \mathbf{a}_t\}_i)$;

2) the Gram matrix $L^f$ which corresponds to variable $\mathbf{s}_{t+1}$ in distribution $P_f$, i.e., $L^f_{ji} = \kappa(\{\mathbf{s}_{t+1}\}_j, \{\mathbf{s}_{t+1}\}_i)$;

3) the Gram matrix $K^c$ which corresponds to concatenated variable $\mathbf{s}_t, \mathbf{a}_t$ in distribution $P_c$;

4) the Gram matrix $L^c$ which corresponds to variable $\mathbf{s}_{t+1}$ in distribution $P_c$;

5) the (cross) Gram matrix $K^{fc}$ (or $K^{cf}$) which measures all pairwise inner products of $\{\mathbf{s}_t, \mathbf{a}_t\}$ from distribution $P_f$ to distribution $P_c$, i.e., $(K^{fc})_{ji} = \kappa(\{\mathbf{s}_t, \mathbf{a}_t\}_j, \{\mathbf{s}_t, \mathbf{a}_t\}_i)$, where $1 \le i \le \tau$ and $\tau + 1 \le j \le 2\tau$;

6) the (cross) Gram matrix $L^{fc}$ (or $K^{cf}$) which measures all pairwise inner products of $\mathbf{s}_{t+1}$ from distribution $P_f$ to distribution $P_c$.

### D.3 DIFFERENCE WITH RESPECT TO (YU ET AL., 2023)

The application of RL described in (Yu et al., 2023) depends on kernel temporal difference (KTD) (Bae et al., 2011) as the underlying RL backbone to produce their results. Kernel Temporal Difference (KTD) represents a straightforward expansion of conventional Q-learning through the incorporation of kernel methods. Yet, it grapples with difficulties in deep reinforcement learning and complex environments. Conversely, our MaxDiv exhibits the versatility to serve as an intrinsic-motivated principle for any RL method.

Additionally, we introduce MaxCondDiv in a principled manner and highlight its connection (Proposition 3) and distinction (thought experiment) from the well-known entropy maximization principle. Within our MaxCondDiv principle, agents are incentivized to explore states that are maximally distant from states in former trajectory portions. However, this characteristic seems to be disproven in (Yu et al., 2023).

This is corroborated by the outcomes observed in the mountain car and maze scenarios. Our MaxDiv results reveal prominent peaks in states located far from the initial points. In sharp contrast, the findings outlined in (Yu et al., 2023) for mountain car and Maze failed to exhibit a similar characteristic.

Based on our experiments using KTD, we discovered that KTD can efficiently explore tabular environments such as Mountain Car and Maze on its own by simply fixing the reward to be zero, without utilizing divergence, and still achieve similar outcomes as shown in (Yu et al., 2023). Consequently, we believe that the exploration in (Yu et al., 2023) is primarily attributed to KTD rather than conditional divergence.

Furthermore, we have also discovered the relationship between CS divergence and KL divergence. We clarify why we choose CS divergence over its KL counterpart using Proposition 1.

In summary, our approach is the first to propose the MaxCondDiv principle, whose prototype idea can be traced back to 1995 (Storck et al., 1995), within the era of deep reinforcement learning. We also first have successfully implemented this principle using popular deep learning methods such as PPO.

### D.4 LIMITAION AND FUTURE WORKS

A limitation of our work is its inapplicability to video-oriented environments such as Atari or Mario. These environments have high-dimensional data with spatial information. However, we believe that this limitation can be addressed in the future by implementing approaches akin to those works centered around the maximum entropy principle, such as employing Variational Autoencoders (VAE) to encode images into latent vectors (Yuan et al., 2022; Zhang et al., 2021).

## E CODES IN PYTORCH

We have included two demo files in the folder: demo_MountainCar.py and demo_thought_exp.py. The demo_MountainCar.py demonstrates the application of our method on the MountainCar environment, while the demo_thought_exp.py replicates our radical exploration in the thought experiments.

We also provide PyTorch structure for CCSD estimation.

```python
def GaussianMatrix(X, Y, sigma):
    size1 = X.size()
    size2 = Y.size()
    G = (X*X).sum(-1)
    H = (Y*Y).sum(-1)
    Q = G.unsqueeze(-1).repeat(1, size2[0])
    R = H.unsqueeze(-1).T.repeat(size1[0], 1)

```

```
 9
10      H = Q + R - 2*X@(Y.T)
11      H = torch.exp(-H/2/sigma**2)
12
13
14      return H
15
16  def CCSD(x1,x2,y1,y2,sigma = 1): # conditional cs divergence
17      x1 = torch.tensor(x1)
18      x2 = torch.tensor(x2)
19      y1 = torch.tensor(y1)
20      y2 = torch.tensor(y2)
21
22
23      K1 = GaussianMatrix(x1,x1,sigma)
24      K2 = GaussianMatrix(x2,x2,sigma)
25
26      L1 = GaussianMatrix(y1,y1,sigma)
27      L2 = GaussianMatrix(y2,y2,sigma)
28
29      K12 = GaussianMatrix(x1,x2,sigma)
30      L12 = GaussianMatrix(y1,y2,sigma)
31
32      K21 = GaussianMatrix(x2,x1,sigma);
33      L21 = GaussianMatrix(y2,y1,sigma);
34
35      H1 = K1*L1
36      self_term1 = (H1.sum(-1)/((K1.sum(-1))**2)).sum(0)
37
38      H2 = K2*L2
39      self_term2 = (H2.sum(-1)/((K2.sum(-1))**2)).sum(0)
40
41      H3 = K12*L12;
42      cross_term1 = (H3.sum(-1)/((K1.sum(-1))*(K12.sum(-1)))).sum(0)
43
44      H4 = K21*L21;
45      cross_term2 = (H4.sum(-1)/((K2.sum(-1))*(K21.sum(-1)))).sum(0)
46
47      cs1 = -2*torch.log2(cross_term1) + torch.log2(self_term1)
48      + torch.log2(self_term2)
49      cs2 = -2*torch.log2(cross_term2) + torch.log2(self_term1)
50      + torch.log2(self_term2)
51
52
53      return ((cs1+cs2)/2).item()
```

