# OpenReview forum: "Reward-Free Exploration by Conditional Divergence Maximization"
_ICLR.cc/2024/Conference — ICLR 2024 Conference Withdrawn Submission_

### Official Review · Reviewer_XX6U · 2023-10-30

**Soundness:** 3 good
**Presentation:** 3 good
**Contribution:** 2 fair
**Rating:** 5
**Confidence:** 4

**Summary:**

This paper introduces a  model-free exploration strategy called Maximum Conditional Divergence (MaxCondDiv) for reinforcement learning. MaxCondDiv leverages the Cauchy-Schwarz (CS) divergence to estimate the disparity in state transition probabilities of two trajectory segments. The author further demonstrates that maximizing the CS divergence corresponds to optimizing the 2nd order Renyi entropy, which establishes a connection to the broader framework of maximum entropy approaches. The empirical evidence presented in this study shows that CS divergence is more stable than KL divergence in the Mujoco environments. Additionally, MaxCondDiv is able to explore more
frequently on far away states from the initial state than a maximum entropy baseline.

**Strengths:**

The author is able to demonstrate clearly the proposed method through figures and explanations. The proposed approach is well-experienced and analyzed with other reward-free exploration strategies.

**Weaknesses:**

While the comparison with KL alternatives and max entropy based methods are well analyzed, there is a question mark on the novelty of the proposed work. In my opinion, the main contribution to the community would be utilizing CS divergence for reward-free exploration (Prop. 2) which looks similar to (Prop. 2) in [Yu et al., 2023].

**Questions:**

I am not sure how relevant experiment 4.1 is on MaxJDiv, especially given the conclusion that maximizing the joint is better than maximizing the conditional but the proposed approach is to maximize the conditional.

How does MaxCondDiv compare to other baselines (like the KL divergence alternatives) in terms of computational cost given the CS divergence computational complexity is O($N^2$)?

Appendix C.5, in  Table 4, what about the KL divergence-based MaxCondDiv? Is it worse or better than CS divergence for downstream tasks?

In the comparison of stability between KL and CS divergence, I would suggest also reporting the variance as the Y axis is different between Figure 12 and 13.

---

### Official Review · Reviewer_hR4c · 2023-11-01

**Soundness:** 1 poor
**Presentation:** 3 good
**Contribution:** 3 good
**Rating:** 3
**Confidence:** 4

**Summary:**

In Reward-Free Exploration by Conditional Divergence Maximization, the author(s) introduce a novel exploration bonus, which they say is based on the “Conditional Cauchy-Schwarz Divergence”. In their implementation, this divergence is between two estimated versions of the state transition distribution, which they estimate in a manner which they refer to as “model-free”. Using a procedure from a 2023 arXiv preprint by Yu et al., estimate the Conditional Cauchy-Schwarz Divergence and train agents in several environments, including a maze environment, MountainCar, and several MuJoCo environments. Their results demonstrate that their method is able to achieve remarkably higher state coverage than other exploration methods with similar goals.

**Strengths:**

This paper has many significant strengths - it introduces a novel measurement of divergence to the literature in Reinforcement Learning and demonstrates very significant gains in a particular measure of exploration, as well as impressive qualitative results. The paper also provides a high-quality review of the existing literature and a very nice introduction to the subject of Rényi alpha-entropy. Further, the paper claims the ability accomplish exploration based on differences in transition probabilities without training an explicit model, which is laudable and could open a range of new options for existing methods.

**Weaknesses:**

This paper has several important weaknesses, but one towers above the others: the central aspect of this work is its estimator of the CS divergence, which is neither introduced in this work nor is it verified in any previous peer-reviewed publication. Without the ability to rely on the claims made about the divergence estimator, it is impossible to recommend this paper for acceptance. The proof provided in that earlier work appears also to be present in the appendix of the presented paper, but this does not appear to be introduced as novel work, and as such I do not review it.

Several points in the paper have unclear and uncited statements, such as 1. “However, the utilization… leading to longer training time”, 2. “they never explicitly estimate the true divergence… in the trajectory”, 3. “our framework estimates an intrinsic reward as defined by the divergence…”, 4. “The estimator exhibits low computational complexity…”, 5. “To maximize the divergence…”. These must be clarified.

At several points in the paper, the notion is invoked that there is some “true” method of estimating transition probabilities based upon different samples of transition triplets. The paper never explains what this “true” method is, nor what the “true” estimate would be, if these are different. This makes the central conceptual argument of the paper somewhat muddled, though the intention is able to break through well and the overall argument would make sense if this could be explained.

**Questions:**

What is meant by the idea, repeated many times in the text and summarized in the abstract: “the divergence between the agent’s estimation of the transition probability between the next state given current state-action pairs … in two adjacent trajectory fractions”? Is a particular method of estimating these assumed, and is there any reason to think that this method would be efficacious? The paper cites a very similar method where this estimation relies on counting (Storck et al., 1995)—that method seems to be on more stable footing, or at least more obviously stable footing. What is the basis in this work for such an estimator? I would like for these questions to be addressed also for the similar matter of “the true divergence of transition probability … from observations … in the trajectory”. This is most severe in section 3.1.

In Section 4.1, the paper states that “To maximize the divergence, the agent needs to move to (100, 102) in the next step.”. This is confusing, as it seems to me that transitioning up (i.e. incrementing the y value) again is more similar to transitioning up from a lower state than transitioning to the right or left (i.e. incrementing or decrementing the x value) would be. Is this claim made on a theoretical or empirical basis?

In Section 4.2, you state that you record trajectories for 50,000 steps in the maze environment, but reset the agent’s position every 1000 steps—that notion of trajectory changes very significantly the notion, discussed earlier in the paper, of basing transition probability estimates on different sections of a trajectory. This procedure is (to my mind, at least) much more akin to basing those estimates on entire trajectories (or, in fact, sets of up to 49 such trajectories). Can you explain this choice?

How many samples were used to generate the figures in the paper? In particular, how many were used for figure 5, and what do the error bars in that figure indicate?

**Details Of Ethics Concerns:**

The authors ask us to review a work whose basis has not itself been subject to peer review, and which they do not ask us to review here. This is not exactly plagiarism or dual submission, but it is harmful and the work obviously is not yet fit for publication. If they had asked us to review that material, it would be a different matter entirely, but that was not the job of the reviewers here, and the authors seem to be attempting to slip their unreviewed theoretical materials past us using an arXiv citation.

---

### Official Review · Reviewer_ovbq · 2023-11-01

**Soundness:** 3 good
**Presentation:** 3 good
**Contribution:** 3 good
**Rating:** 5
**Confidence:** 3

**Summary:**

This work proposes a new curiosity-driven exploration method called maximum conditional divergence (MaxCondDiv). They define curiosity as trajectory-wise prediction errors. In addition to some theoretical analysis of the method, some experiments show that MaxCondDiv outperforms other baselines to a degree.

**Strengths:**

(1) The method is sound and valuable.

(2) The mathematical analysis is correct.

(3) The modeling of the conditional distributions makes learning more efficient without trained dynamics models.

**Weaknesses:**

(1) The authors claim that "it reduces internal model selection bias". However, there are no experiments or visualizations towards this conclusion.

(2) The experiments are not sufficient to support the SoTA results, as some hard exploration tasks are not chosen, such as  Montezuma in atari and DMControl suite (not OpenAI gym, which is much easier).

**Questions:**

(1) How this method is sensitive to $\tau$? (about the computational costs and performance)

(2) If training the dynamics function to model the distributions, how would MaxCondDiv perform? No ablations for this, and how less computationally expensive is the MaxCondDiv?

---

### Official Review · Reviewer_3dXY · 2023-11-01

**Soundness:** 2 fair
**Presentation:** 2 fair
**Contribution:** 2 fair
**Rating:** 3
**Confidence:** 4

**Summary:**

The paper focusses on developing a model-free curiosity-driven exploration in reward-free environments by maximizing an intrinsic bonus derived from the Cauchy-Schwarz divergence between the model estimation of two adjacent trajectory segments. The performance is validated with improved performance on several Mujoco benchmarks and shows to learn new skills in reward-free settings.

**Strengths:**

Intrinsic curiosity-driven rewards to learn under sparse rewards have been leveraged in several past research and one of the primary novelties of the work lies in formulating curiosity in a model-free manner, which is computationally less extensive than the model-based as in several prior research. The research also introduces condition CS divergence as a reward function b/w the transition of adjacent trajectory segments which is interesting, unlike prior methods primarily relying on f-divergences. The experimental analysis shows improvement over baselines for reward-free exploration.

**Weaknesses:**

The paper proposes an interesting heuristic in a reward-free exploration scenario, but the mathematical formulation is not extremely clear and needs more clarity. It’s not explicitly clear why the formulation in (9) can probably lead to better exploration which has been explained intuitively with experiments. For example, the recent work in [1] has shown mutual information as a provable source of exploration and demonstrated improved performance under sparsity with information-directed regret analysis. However, current work lacks any such metric to characterize the exploration induced by the divergence. Additionally, the computational tractability of Cauchy Schwartz divergence is not very clear which requires the computation of several Gram matrices resulting in O(n^2) complexity, which is marginally better than modeling the dynamics with Gaussian process which requires a complexity of O(n^3) which can be improved easily with sparse GP or compression[2].  Then the model-free curiosity with less computational complexity is not very appropriate. Hence to clear it, will be helpful to discuss non-parametric transition estimation methods and this approach. The most important aspect is that with GPs, we can estimate exact epistemic uncertainty (which provably induces exploration as intrinsic rewards, since is connected to information gain), however how the proposed CS divergence can help and a precise connection to epistemic uncertainty is not clear (Equation 11 is not sufficient for that or needs more explaination). Additionally, it has been shown in several recent works that GPs [2, 3, 4] or ensembles are sufficient in learning dynamics in Mujoco tasks. Hence, will be helpful to provide a discussion around the same and compare tasks with more complex dynamics to understand the effect.

1. Botao Hao, Tor Lattimore, Regret Bounds for Information-Directed Reinforcement Learning, https://arxiv.org/abs/2206.04640
2. Souradip Chakraborty, Amrit Singh Bedi, Alec Koppel, Brian M. Sadler, Furong Huang, Pratap Tokekar, Dinesh Manocha Posterior Coreset Construction with Kernelized Stein Discrepancy for Model-Based Reinforcement Learning https://arxiv.org/abs/2206.01162
3. Ying Fan, Yifei Ming Model-based Reinforcement Learning for Continuous Control with Posterior Sampling https://arxiv.org/abs/2012.09613
4. Kurtland Chua, Roberto Calandra, Rowan McAllister, Sergey Levine  Deep Reinforcement Learning in a Handful of Trials using Probabilistic Dynamics Models https://arxiv.org/abs/1805.1211

**Questions:**

1. Why MaxDiv learns Skill without Using Extrinsic Rewards is not extremely clear and will be helpful to provide a detailed explanation
2. Currently it focuses on max entropy objectives which is nice, but the validation of reward-free exploration in certain downstream tasks or some sparse reward tasks will be helpful to understand the efficacy of the proposed exploration method, as well as in some prior research.
3. How does the proposed method compare with scenarios when we predict the dynamics with GPs and leverage the posterior uncertainty be helpful? A comparison with PSRL or Thompson sampling and information-directed approaches in the context will be helpful [1].



[1]. Christoph Dann, Mehryar Mohri, Tong Zhang, Julian Zimmert A Provably Efficient Model-Free Posterior Sampling Method for Episodic Reinforcement Learning https://arxiv.org/abs/2208.10904